# Continual learning via probabilistic exchangeable sequence modelling

**Hanwen Xing**  *hanwen.xing@wrh.ox.ac.uk*
*Nuffield Department of Women's and Reproductive Health*
*University of Oxford*

**Christopher Yau**  *christopher.yau@wrh.ox.ac.uk*
*Nuffield Department of Women's and Reproductive Health*
*University of Oxford &*
*Health Data Research UK*

**Reviewed on OpenReview:** *https://openreview.net/forum?id=fDnAsRUkOF*

## Abstract

Continual learning (CL) refers to the ability to continuously learn and accumulate new knowledge while retaining useful information from past experiences. Although numerous CL methods have been proposed in recent years, it is not straightforward to deploy them directly to real-world decision-making problems due to their computational cost and lack of uncertainty quantification. To address these issues, we propose CL-BRUNO, a probabilistic, Neural Process-based CL model that performs scalable and tractable Bayesian update and prediction via probabilistic exchangeable sequence modelling. Our proposed approach uses deep-generative models to create a unified Bayesian probabilistic framework capable of handling different types of CL problems such as task- and class-incremental learning by modelling data from different tasks as sequences of exchangeable random variables, allowing users to integrate information across different CL scenarios efficiently using a single model, and give easy-to-interpret probabilistic predictions without the need of training or maintaining separate classifiers. Our approach is able to prevent catastrophic forgetting through distributional and functional regularisation without the need of retaining any previously seen samples, making it appealing to applications where data privacy or storage capacity is of concern. Experiments show that CL-BRUNO outperforms existing methods on both natural image and biomedical data sets, confirming its effectiveness in real-world applications.

## 1 Introduction

Continual learning (CL) enables an intelligent system to develop and refine itself adaptively in accordance with real-world dynamics by incrementally accumulating and exploiting knowledge gained from previous experience without the need to train any new model from scratch (Hassabis et al., 2017). The main challenge CL has to tackle is known as *catastrophic forgetting*, which refers to previously learned knowledge being drastically interfered by new information (McClelland et al., 1995; McCloskey and Cohen, 1989). In order to deliver accurate and trustworthy predictions, a CL model in practice should, on one hand, be able to integrate new knowledge efficiently based on the stream of new inputs from dynamic data distributions (learning plasticity) and, on the other hand, maximally retain relevant information from the past and prevent catastrophic forgetting (memory stability). The competition between these two conflicting objectives is known as *stability-plasticity dilemma*, which has been widely studied from both biological and computational perspectives (Ditzler et al., 2015; Parisi et al., 2019).

Numerous methods and strategies have been developed to address the CL problem under different scenarios (Wang et al., 2024): Regularisation-based methods (Kirkpatrick et al., 2017; Schwarz et al., 2018; Wang et al.,

2021) aim to retain knowledge from history by introducing regularisation terms to balance the old and new tasks. Such regularisation can either be at a parameter level, e.g. minimising changes to key model parameters (under some data-driven parameter-wise importance measure) as new tasks are being learnt (Aljundi et al., 2018; Zenke et al., 2017), or at a functional level using e.g. knowledge distillation to prevent the model's performance on previous tasks from drastic deterioration (Michel et al., 2024; Rudner et al., 2022; Titsias et al., 2019). Architecture-based CL approaches (Thapa and Li, 2024; Gurbuz and Dovrolis, 2022; Mallya et al., 2018; Dhar et al., 2019) seek to mitigate catastrophic forgetting and inter-task interference by allocating specific sets of parameters to different tasks. Wortsman et al. (2020); Xue et al. (2022); Kang et al. (2022) use binary masks to select and isolate the subset of dedicated parameters for different tasks in a fixed-size model. Hung et al. (2019); Draelos et al. (2017) accommodate additional tasks by dynamically expanding the model architecture, providing additional model capacity before the model "saturates" as more incremental tasks are introduced. Another popular approach is rehearsal-based methods (Robins, 1995) where historical information is retained by approximating and recovering historical data distributions. Wen et al. (2024); Shim et al. (2021); Aljundi et al. (2019); Rebuffi et al. (2017) retain distributional information by selecting and storing representative samples from the old data. Despite its conceptual simplicity, such approaches known as *experience replay* can be infeasible due to storage or data privacy constraints. *Generative replay* methods (Petit et al., 2023; Chen et al., 2022; Egorov et al., 2021; Shin et al., 2017) address this issue by summarising old data distributions as generative models, and replay generated data instead of the actual samples.

In practice, experience-replay methods such as Jha et al. (2024) and Wen et al. (2024) are computationally intensive and require storing historical data as part of their memory states, which could be infeasible in many applications due to privacy or storage concerns. Generative (Petit et al., 2023; Gopalakrishnan et al., 2022; Wu et al., 2018) and regularisation-based CL methods (Dohare et al., 2024; He and Zhu, 2022; Li and Hoiem, 2017) avoid the need of storing historical data, but are also computational costly and lack statistically principled updating or uncertainty quantification schemes. These limitations make it difficult to deploy them directly to real-world problems. To address these limitations, we propose Continual Learning Bayesian RecUrrent Neural mOdel (**CL-BRUNO**), a probabilistic, Neural Process-based CL model that performs scalable and tractable Bayesian update and prediction. CL-BRUNO utilises deep generative models, and extends the exchangeable sequence modelling framework given by Korshunova et al. (2020) (Conditional-BRUNO) to provide a versatile probabilistic framework capable of performing probabilistic label and task-identity estimation, and handling different CL scenarios such as task- and class-incremental learning using a common, likelihood-based updating scheme. This means that users can handle different types of CL problems using a single CL-BRUNO model in a statistically principled fashion, allowing knowledge to be accumulated more efficiently. Our proposed method uses *generative replay* (Shin et al., 2017; Wu et al., 2018) as a regularisation to prevent catastrophic forgetting without the need of retaining previously seen samples, making it more appealing in real-world applications where data privacy or storage capacity is of concern. Numerical experiments show that CL-BRUNO outperforms existing methods on both natural image and biomedical datasets, confirming its effectiveness. This paper is structured as follows: We first give technical background in Sec 2, then describe the proposed CL-BRUNO model in Sec 3. We highlight its connection with existing works in Sec 4, and report numerical experiments in Sec 5[1]. We conclude the paper with a brief discussion.

## 2 Background

In this section, we give the technical background relevant to our proposed method.

### 2.1 Normalising flow

A discrete Normalising flow (NF) (Rezende and Mohamed, 2015; Dinh et al., 2016; Papamakarios et al., 2017) models a continuous probability distribution by transforming a simple-structured base distribution (e.g. isotropic multivariate Normal) to the more complex target using a bijective transformation $T$ parameterised as a composition of a series of smooth and invertible mappings $f_1, ..., f_K$ with easy-to-compute Jacobians. This $T$ is applied to the "base" random variable $z_0 \sim p_0$, where $z_0 \in \mathbb{R}^D$ and $p_0$ is the known base density.

---

[1]Code reproducing the reported results can be found in https://github.com/hwxing2357/cl_bruno.

Let $z_k = f_k \circ f_{k-1} \circ ... \circ f_1(z_0)$ for $k = 1, ..., K$. By applying change of variable repeatedly, the final output $z_K$ has density $p_K(z_K) = p_0(z_0) \prod_{k=1}^{K} |\det J_k(z_{k-1})|^{-1}$, where $J_k$ is the Jacobian of the mapping $f_k$. The final density $p_K$ can be used to approximate target distributions with complex structure, and one can sample from $p_K$ easily by applying $T = f_K \circ f_{K-1} \circ ... \circ f_1$ to $z_0 \sim p_0$. In order to evaluate $p_K$ efficiently, we are restricted to transformations $f_k$ whose $\det J_k(z)$ is easy to compute. For example, Real-NVP (Dinh et al., 2016) uses the following family of transformations: For $m \in \mathbb{N}$ such that $1 < m < d$, let $z_{1:m}$ be the first $m$ entries of $z \in \mathbb{R}^D$, let $\times$ be element-wise multiplication and let $\mu_k, \sigma_k : \mathbb{R}^m \to \mathbb{R}^{D-m}$ be two neural nets. The smooth and invertible transformation $y = f_k(z)$ for each step $k$ in a Real-NVP is defined as

$$y_{1:m} = z_{1:m}, \quad y_{m+1:d} = \mu_k(z_{1:m}) + \sigma_k(z_{1:m}) \times z_{m+1:d} \tag{1}$$

The Jacobian $J_k$ of $f_k$ is lower triangular, hence $\det J_k(z) = \prod_{i=1}^{D-m} \sigma_{ik}(z_{1:m})$, where $\sigma_{ik}(z_{1:m})$ is the $i$th entry of $\sigma_k(z_{1:m})$. Continuous normalising flows (Chen et al., 2018; Onken et al., 2021; Lipman et al., 2023) further extend model flexibility by replacing the series of transformations by a vector field continuously indexed by pseudo-time.

## 2.2 Conditional-BRUNO

Korshunova et al. (2020) proposed Conditional Bayesian Recurrent Neural model (C-BRUNO), a Neural Process (Garnelo et al., 2018; Kim et al., 2019; Xu et al., 2024) which models exchangeable sequences of high-dimensional observations conditionally on a set of labels. Given a set of feature-label pairs $\{X_i, y_i\}_{i=1}^{N}$ where $X_i \in \mathbb{R}^D$ is a feature vector and $y_i$ its corresponding label, C-BRUNO assumes that the joint distribution $p(X_1, ..., X_N | y_1, ..., y_N) = p(X_{\pi(1)}, ..., X_{\pi(N)} | y_{\pi(1)}, ..., y_{\pi(N)})$ for any permutation $\pi$. In other words, this permutation invariance implies that the model is agnostic to the ordering of the pairs $\{X_i, y_i\}$. Therefore, the sequence is exchangeable as any reordering of the sequence does not change the joint distribution $p(X_1, ..., X_N | y_1, ..., y_N)$. C-BRUNO models $p(X_1, ..., X_N | y_1, ..., y_N)$ by first transforming each feature vector $X_i$ into a latent variable $\mathbf{z}_i = f_\theta(X_i; y_i)$, where $f_\theta(X; y) : \mathbb{R}^D \to \mathbb{R}^D$ is a bijective conditional Real-NVP depending on label $y$. Denote $z_i^{(d)}$ as the $d$th entry of $\mathbf{z}_i$. It then models each dimension $d$ of the exchangeable latent sequence $\{\mathbf{z}_i\}_{i=1}^{N}$ as an independent 1-D Gaussian process with $\mathrm{Var}(z_i^{(d)}) = \nu^{(d)}$ and $\mathrm{Cov}(z_i^{(d)}, z_j^{(d)}) = \rho^{(d)}$ for all $i, j = 1, ..., N$, where $0 < \rho^{(d)} < \nu^{(d)}$ are trainable covariance parameters (Note that we need $0 < \rho^{(d)} < \nu^{(d)}$ to ensure positive definiteness of the covariance matrix). In other words, C-BRUNO assumes that $p(\mathbf{z}_1, ..., \mathbf{z}_N) = \prod_{d=1}^{D} p_d(\{z_n^{(d)}\}_{n=1}^{N})$ where $p_d(\{z_n^{(d)}\}_{n=1}^{N}) = \mathcal{N}(\{z_n^{(d)}\}_{n=1}^{N}; \mathbf{0}_N, \Sigma_d)$, $\mathbf{0}_N$ is a zero vector of length $N$, and $\Sigma_d$ is a $N \times N$ covariance matrix with diagonal element $\nu^{(d)}$ and off-diagonal element $\rho^{(d)}$. Suppose the bijective transformation $f_\theta$ and the covariance parameters $\{\rho^{(d)}, \nu^{(d)}\}_{d=1}^{D}$ have been chosen. C-BRUNO generates a new sample from the predictive distribution $p(X^*|y^*, X_{1:N}, y_{1:N})$ conditioned on new label $y^*$ and previously seen feature-label pairs by first sampling $\mathbf{z}^* \sim p(\mathbf{z}^*|\mathbf{z}_{1:N})$ and then computing $X^* = f_\theta^{-1}(\mathbf{z}^*; y^*)$. The predictive likelihood $p(X_{N+1}|y_{N+1}, X_{1:N}, y_{1:N})$ of a new pair $(X_{N+1}, y_{N+1})$ can be evaluated in a similar fashion. Thanks to the specific covariance function used in C-BRUNO, sampling from and evaluating $p(\mathbf{z}^*|\mathbf{z}_{1:N})$ using the recursive formula given in Korshunova et al. (2020) has a time complexity $\mathcal{O}(N)$ instead of the general case $\mathcal{O}(N^3)$. This greatly improves the scalability of C-BRUNO as it scales linearly with both sample size $N$ and data dimension $D$.

Our proposed CL-BRUNO is built on the C-BRUNO framework: The exchangeability assumption in C-BRUNO imposes a permutation-invariance architecture to the model. It includes the conditionally identically and independently distributed (conditionally i.i.d.) assumption as a special case, therefore provides a more flexible modelling framework to reason about future samples based on the observed ones with minimal additional computational cost, and has been proven useful in e.g. few-shot learning and anomaly detection (Korshunova et al., 2020). This feature is appealing in many CL scenarios where new datasets arrive incrementally in batches: In comparison with generative-replay methods based on i.i.d. assumption, e.g. Scardapane et al. (2020), modelling samples in each dataset as an exchangeable sequence using C-BRUNO allows users to better capture and aggregate distributional information such as sample sizes and inter-sample correlations. To see that, note C-BRUNO's Gaussian process predictive base distribution $p(\mathbf{z}^*|\mathbf{z}_{1:N})$ is updated as the sample size $N$ increases, while models based on the i.i.d. assumption, e.g. Scardapane et al. (2020), ignore this dependency between $\mathbf{z}^*$ and $\mathbf{z}_{1:N}$, and set $p(\mathbf{z}^*|\mathbf{z}_{1:N}) = p(\mathbf{z}^*)$ to be some user-specified distribution such as an isotropic Gaussian. This aggregation feature is useful for probabilistic prediction and

inference in a natural and statistically principled fashion. Furthermore, our proposed CL-BRUNO leverages the efficient and exact density evaluation feature of C-BRUNO to directly estimate probability distributions of both class and task identity of any test data (Sec 3.2). This means that compared with existing methods such as Petit et al. (2023) and Wu et al. (2018), our proposed method does not require training or maintaining separate neural network based classifiers, making it computationally more efficient. Additionally, the output of CL-BRUNO can be interpreted as approximate Bayesian classifiers, which is more interpretable than the logit scores given by a neural network based classifier. We give details of the proposed CL-BRUNO in the following section.

## 3 Method

In this section, we give technical details of our proposed method. We start by specifying the notation. Suppose there is a sequence of tasks labelled by $t = 1, 2, \ldots, T$. Assuming all tasks are classification problems with potentially different or disjoint label spaces (extensions to regression problems are straightforward). We suppose each task $t = 1, 2, \ldots, T$ is associated with dataset $\mathcal{D}_t = \{\mathcal{X}_t, \mathcal{Y}_t\}$ where $\mathcal{X}_t = \{X_{i,t}\}_{i=1}^{N_t}$ is the feature set, $X_{i,t} \in \mathbb{R}^d$ is the feature vector of the $i$-th sample in the $t$-th task, $\mathcal{Y}_t = \{Y_{i,t}\}_{i=1}^{N_t}$ is the label set, $Y_{i,t} \in \{1, ..., C_t\}$ is the $i$-th sample's corresponding label, and $N_t, C_t$ are the sample size of $\mathcal{D}_t$ and the number of distinct labels in the $t$-th task respectively. Similar to Korshunova et al. (2020), for each dataset $\mathcal{D}_t = \{\mathcal{X}_t, \mathcal{Y}_t\}$, we assume that (a) the corresponding density function $p_t(\mathcal{X}_t, \mathcal{Y}_t)$ factorises as

$$p_t(\mathcal{X}_t, \mathcal{Y}_t) = p(\mathcal{X}_t | t, \mathcal{Y}_t) p(\mathcal{Y}_t | t) = p(X_{1,t}, ..., X_{N_t,t} | t, Y_{1,t}, ..., Y_{N_t,t}) \prod_{i=1}^{N_t} p(Y_{i,t} | t), \tag{2}$$

i.e. $\mathcal{Y}_t$ are generated i.i.d. from some task-specific label prior $p(\cdot|t)$ independent to the generative process of $\mathcal{X}_t$, and (b) $\mathcal{X}_t$ is an exchangeable sequence of feature vectors conditioned on both task identity $t$ and label set $\mathcal{Y}_t$, i.e.

$$p(X_{1,t}, ..., X_{N_t,t} | t, Y_{1,t}, ..., Y_{N_t,t}) = p(X_{\pi(1),t}, ..., X_{\pi(N_t),t} | t, Y_{\pi(1),t}, ..., Y_{\pi(N_t),t}) \tag{3}$$

for any permutation $\pi$ of size $N_t$.

We propose Continual Learning BRUNO (CL-BRUNO) under the distributional assumptions given above. The key motivation of CL-BRUNO is to formulate continual learning problems as modelling streams of data (one for each task), which may come in batches, as a collection of exchangeable sequences. From this perspective, once $N$ samples from a stream of data (i.e. a continual learning task) have been observed, users can then extract distributional information from historical observations by modelling the joint distribution $p(X_{1:N,t}, Y_{1:N,t})$, and predict a new sample and/or the associated label by inferring the one-step-ahead conditional distribution $p(X_{N+1,t}, Y_{N+1,t} | X_{1:N,t}, Y_{1:N,t})$ or $p(Y_{N+1,t} | X_{1:N+1,t}, Y_{1:N,t})$. Specifically, CL-BRUNO consists of two modules: a C-BRUNO model that models feature vectors $\mathcal{X}_t$ from different task $t$ as exchangeable sequences, and an approximate Bayesian inference pipeline for label and task identity estimation. Let $\hat{p}(\mathcal{X}_t | t, \mathcal{Y}_t)$ be a C-BRUNO model that approximates $p(\mathcal{X}_t | t, \mathcal{Y}_t)$, the true conditional distribution of feature vectors $\mathcal{X}_t$ given $(t, \mathcal{Y}_t)$. CL-BRUNO aims to train a collection of C-BRUNO models $\{\hat{p}(\mathcal{X}_t | t, \mathcal{Y}_t)\}_{t=1,2,\ldots}$ incrementally over an expanding range of $(t, \mathcal{Y}_t)$ by continually learning generative models $\hat{p}(\mathcal{X}_{t'} | t', \mathcal{Y}_{t'})$ from new datasets $\mathcal{D}_{t'}$, while preventing those it has learnt from previous tasks from being interfered or disrupted: If the incrementally trained C-BRUNO models could give $\hat{p}(\mathcal{X}_t | t, \mathcal{Y}_t)$ that well approximates $p(\mathcal{X}_t | t, \mathcal{Y}_t)$ for all previously seen tasks $t$ throughout the continual learning process, then users could handle queries such as estimating the task identity associated with a new feature vector $X^*$, or computing $\hat{p}(Y^* | X^*, t, \mathcal{D}_t)$, the approximate posterior distribution of label $Y^*$ associated with a new $X^*$ from the $t$-th task based on historical data at any stage of the continual learning process in a statistically principled fashion.

### 3.1 Incremental learning scheme

In this section, we discuss the parametrisation and incremental training procedure of CL-BRUNO. Thanks to the i.i.d. assumption on labels $\{\mathcal{Y}_t\}_{t=1}^T$ in 2, the task-specific label priors $p(Y|t)$ can be estimated directly using their population proportions. For the remainder of the section, we focus on learning the conditional distribution of $\mathcal{X}_t$. Specifically, the C-BRUNO model used in our proposed method consists of a Conditional

Real-NVP[2] $f_\theta(\cdot; t, Y_t) : \mathbb{R}^D \to \mathbb{R}^D$ parameterised by $\theta$ as a bijective transformation that depends on both task identity $t$ and label $Y_t$. We parameterise each task identity $t$ and each of the unique labels $\{1, \ldots, C_t\}$ in the $t$-th task as trainable embeddings $r_t, s_{j,t} \in \mathbb{R}^l$ respectively for $j = 1, \ldots, C_t$ and $t = 1, 2, \ldots$. For the rest of the paper, the embeddings $\{r_t, \{s_{j,t}\}_{j=1}^{C_t}\}_{t=1}^T$ are viewed as parts of the Conditional Real-NVP parameter $\theta$. For $t = 1, 2, \ldots$, let $\mathbf{z}_{i,t} = f_\theta(X_{i,t}; t, Y_{i,t})$ for all $i = 1, \ldots, N_t$, $\mathbf{z}_t = \{\mathbf{z}_{i,t}\}_{i=1}^{N_t}$, and $p_{\lambda_t}(\mathbf{z}_t)$ be the task-specific latent distribution parametrised by $\lambda_t = \{\nu_t^{(d)}, \rho_t^{(d)}\}_{d=1}^D$ as described in Sec 2.2. Denote $\boldsymbol{\lambda} = \{\lambda_t\}_{t=1,2,\ldots}$ the set of latent distribution parameters. With a slight abuse of notation, we see $X_{1:0,t}$, $Y_{1:0,t}$ and $\mathbf{z}_{1:0,t}$ as empty sets for the rest of the paper.

**Learning from scratch**

If there is no historical information or previously seen data, then we can extract and retain information in these datasets by training a series of C-BRUNO models that approximate $p(\mathcal{X}_t | t, \mathcal{Y}_t)$ for all $t = 1, \ldots, T$ by minimising the negative log likelihood

$$
\begin{aligned}
L(\theta, \boldsymbol{\lambda}; \{\mathcal{D}_t\}_{t=1}^T) &= -\sum_{t=1}^T \overbrace{\log p_{\theta, \lambda_t}(\mathcal{X}_t | t, \mathcal{Y}_t)}^{\text{task-wise log likelihood}} \\
&= -\sum_{t=1}^T \sum_{i=1}^{N_t} \log p_{\theta, \lambda_t}(X_{i,t} | Y_{i,t}, t, X_{1:i-1,t}, Y_{1:i-1,t}) \\
&= -\sum_{t,i=1}^{T,N_t} \log \left( p_{\lambda_t}(\mathbf{z}_{i,t} | \mathbf{z}_{1:i-1,t}) \left| \det \frac{\partial f_\theta(X_{i,t}; t, Y_{i,t})}{\partial X_{i,t}} \right| \right),
\end{aligned}
\tag{4}
$$

as suggested by Korshunova et al. (2020) where $p_{\theta, \lambda_t}(\mathcal{X}_t | t, \mathcal{Y}_t)$ denotes the C-BRUNO approximation to $p(\mathcal{X}_t | \mathcal{Y}_t)$. Now, suppose $(\hat{\theta}, \hat{\boldsymbol{\lambda}}) = \arg\min_{\theta, \boldsymbol{\lambda}} L(\theta, \boldsymbol{\lambda}; \{\mathcal{D}_t\}_{t=1}^T)$, and $\hat{\mathbf{z}}_{i,t} = f_{\hat{\theta}}(X_{i,t}; t, Y_{i,t})$ for all $i = 1, \ldots, N_t$ and $t = 1, \ldots, T$. Then for any task $t = 1, \ldots, T$, $p(X^* | Y^*, t, \mathcal{D}_t)$, the predictive distribution of new observation $X^*$ given label $Y^*$ and historical data $\mathcal{D}_t$, can be summarised by a generative C-BRUNO model with bijective mapping $f_{\hat{\theta}}$ and multivariate Gaussian predictive distribution $p_{\hat{\lambda}_t}(\cdot | \hat{\mathbf{z}}_{1:N_t,t})$. See also Fig 1 (a) for a schematic illustration.

We stress that even though $p_{\hat{\lambda}_t^{(T)}}(\cdot | \hat{\mathbf{z}}_{1:N_t,t})$ is a multivariate Gaussian distribution whose mean and covariance depend on $\hat{\mathbf{z}}_{1:N_t,t}$, evaluating or sampling from it does not require access to $\hat{\mathbf{z}}_{1:N_t,t}$, see also Eqn (7) in Korshunova et al. (2020). As a result, once the generative model has been trained, users can use it as a proxy of $\{\mathcal{D}_t\}_{t=1}^T$, and handle queries without the need of revisiting any historical data. This learning strategy allows users to retain knowledge and information from different tasks without storing any sample from any dataset. However, under a CL setup, such a learning scheme via joint likelihood maximisation is not feasible as datasets $\mathcal{D}_t$s are observed in a sequential fashion, and we do not necessarily have access to any previously seen datasets. On the other hand, sequentially maximising $L(\theta, \boldsymbol{\lambda}; \mathcal{D}_t), t = 1, 2, \ldots$ naively whenever a new dataset $\mathcal{D}_t$ arrives will drastically disrupt what it has learnt from historical data due to *catastrophic forgetting* (McCloskey and Cohen, 1989), rendering it unusable for any prediction task except for the most recent one. To address this issue, our proposed CL-BRUNO uses *generative replay* to prevent catastrophic forgetting through both distributional and functional regularisation. We demonstrate our incremental learning strategy under two common CL scenarios known as Task- and Class-Incremental learning. Other scenarios such as Domain- or Instance- Incremental learning (Wang et al., 2024) can be handled in a similar fashion.

**Task-incremental learning**

Suppose our CL model has been trained on $T$ tasks. Let $f_{\hat{\theta}^{(T)}}$, $\{p_{\hat{\lambda}_t^{(T)}}(\cdot | \hat{\mathbf{z}}_{1:N_t,t})\}_{t=1}^T$ be the trained normalising flow and predictive latent distributions respectively. Suppose now a new task $T + 1$ and the associated dataset $\mathcal{D}_{T+1}$ arrive. Our goal is to update the generative model so that it adapts to $\mathcal{D}_{T+1}$ while retaining

---

[2]It is straightforward to replace it with more sophisticated alternatives (Chen et al., 2018; Lipman et al., 2023; Shi et al., 2024). We do not consider them here for sake of conceptual simplicity.

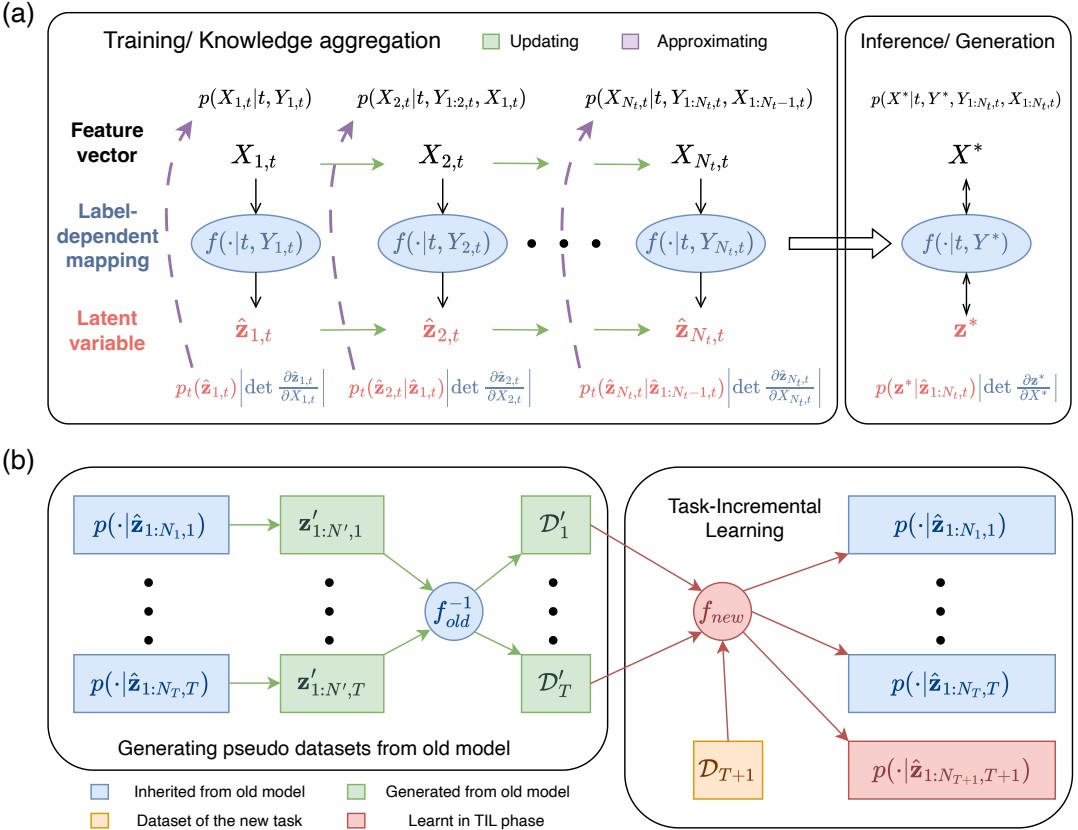

Figure 1: **(a) Schematic illustration of how C-BRUNO learns the sequence distribution** $p(\mathcal{X}_t|\mathcal{Y}_t) = \prod_{i=1}^N (X_{i,t}|X_{1:i-1,t}, Y_{1:i,t})$. For each feature vector $X_{i,t}$ in the sequence, C-BRUNO first transform it to the corresponding latent variable using the label-dependent mapping $\hat{\mathbf{z}}_{i,t} = f(X_{i,t}|Y_{i,t})$, then approximates the one-step-ahead conditional $p(X_{i,t}|X_{1:i-1,t}, Y_{1:i,t})$ by $p_t(\hat{\mathbf{z}}_{i,t}|\hat{\mathbf{z}}_{1:i-1}) \left|\det \frac{\partial \hat{\mathbf{z}}_{i,t}}{\partial X_{i,t}}\right|$. Exchangeability is guaranteed by the specific covariance function in the latent distribution $p(\hat{\mathbf{z}}_{1:N_t})$. In the generation/inference phase, given a label $Y^*$, a new latent variable $\mathbf{z}^*$ is first generated from $p(\cdot|\hat{\mathbf{z}}_{1:N_t})$, a multivariate Gaussian whose mean and covariance depend on the observed sequence, and then transformed to the generated feature vector $X^* = f^{-1}(\mathbf{z}^*|t, Y^*)$ under label $Y^*$. **(b) Schematic illustration of TIL in CL-BRUNO.** Pseudo datasets are generated from the previous latent predictive distributions $p(\cdot|\hat{\mathbf{z}}_{1:N_t,t})$ and the bijective mapping $f_{old}$. Note that in the TIL phase, the new bijective mapping $f_{new}$ learns to 1) map the new dataset $\mathcal{D}_{T+1}$ to a series of latent variables and compute the corresponding latent predictive $p(\cdot|\hat{\mathbf{z}}_{1:N_{T+1},T+1})$ (i.e. learning from new data) and 2) map the pseudo-datasets $\hat{\mathcal{D}}_t$ back to latent variables that resemble samples drawn from the previous latent predictive distributions $p(\cdot|\hat{\mathbf{z}}_{1:N_t,t})$ (i.e. retaining learnt knowledge).

the historical knowledge the old generative model has learnt regarding the previous $T$ tasks. This scenario is known as *Task-Incremental learning* (TIL) (Van de Ven and Tolias, 2019).

To achieve this goal, we update the old CL-BRUNO by estimating 1) a new bijective transformation $f_\theta$ parametrised by $\theta$, and 2) $\lambda_{T+1}$ for the latent distribution of the $T+1$-th task. The rest of the old CL-BRUNO model (i.e. latent distributions associated with previous tasks) are kept unchanged. Specifically, we first generate pseudo datasets $\mathcal{D}'_t = \{X'_{j,t}, Y'_{j,t}\}_{j=1}^{N'}$ for each previously seen task $t = 1, 2, \ldots, T$ from the old CL-BRUNO model as follows. Denote $N'$ the size of pseudo datasets, for each $\mathcal{D}'_t$ and $j = 1, \ldots, N'$, the pseudo label $Y'_{j,t}$ associated with the $t$-th task is sampled according to the population label proportion of $\mathcal{D}_t$, then the pseudo feature vector $X'_{j,t}$ is generated by first drawing latent variable $\mathbf{z}'_{j,t} \sim p_{\hat{\lambda}_t}(\cdot|\hat{\mathbf{z}}_{1:N_t,t})$, then set $X'_{j,t} = f_{\hat{\theta}(T)}(\mathbf{z}'_{j,t}; t, Y'_{j,t})$. Once the $T$ pseudo datasets based on the old CL-BRUNO has been generated, one

can then update the CL-BRUNO by updating parameters $\{\theta, \lambda_{T+1}\}$ using the joint likelihood optimisation approach in Eq 4 based on the augmented dataset $\{\mathcal{D}'_t\}_{t=1}^T \cup \mathcal{D}_{T+1}$. See Fig 1 (b) for a schematic illustration. To further prevent the historical knowledge in $f_{\hat{\theta}(T)}$ from being disrupted by the new dataset $\mathcal{D}_{T+1}$, we adopt the functional regularisation technique in Wu et al. (2018) that penalises the $L_2$ distance between outputs of the old and new models generated from the same set of input noises. Specifically, for a new bijective $f_\theta$ parametrised by $\theta$, the regularisation takes the form

$$R(\theta; \hat{\theta}^{(T)}, \{\mathcal{D}'_t\}_{t=1}^T) = \sum_{t,i=1}^{T,N'} ||f_\theta^{-1}(\mathbf{z}'_{i,t}; t, Y'_{i,t}) - X'_{i,t}||_2^2, \tag{5}$$

where $f_\theta^{-1}(\mathbf{z}'_{i,t}; t, Y'_{i,t})$ and $X'_{i,t}$ are the outputs of the new and old normalising flows based on the common noise $\mathbf{z}'_{i,t}$. Combining the augmented datasets and the functional regularisation, the new generative model $f_\theta$ and the base distribution parameter $\lambda_{T+1}$ of the new task $T+1$ under the TIL scenario is chosen by solving $\min_{\theta, \lambda_{T+1}} L_{\text{TIL}}(\theta, \lambda_{T+1})$ where $L_{\text{TIL}}(\theta, \lambda_{T+1})$ is the regularised joint negative log likelihood

$$L_{\text{TIL}}(\theta, \lambda_{T+1}) = \overbrace{L(\theta, \lambda_{T+1}; \mathcal{D}_{T+1})}^{\text{neg log likelihood of new task}} + \underbrace{\alpha_1 L'(\theta, \{\hat{\lambda}_t^{(T)}\}_{t=1}^T; \{\mathcal{D}'_t\}_{t=1}^T)}_{\text{distributional regulariser}} + \underbrace{\alpha_2 R(\theta; \hat{\theta}^{(T)}, \{\mathcal{D}'_t\}_{t=1}^T)}_{\text{functional regulariser}}, \tag{6}$$

where

$$L'(\theta, \{\hat{\lambda}_t^{(T)}\}_{t=1}^T; \{\mathcal{D}'_t\}_{t=1}^T) = -\sum_{t,j=1}^{T,N'} \log\left[p_{\hat{\lambda}_t^{(T)}}(f_\theta(X'_{j,t}; t, Y'_{j,t})|\hat{\mathbf{z}}_{1:N_t,t})\right] - \sum_{t,j=1}^{T,N'} \log\left[\left|\det \frac{\partial f_\theta(X'_{j,t}; t, Y'_{j,t})}{\partial X'_{i,t}}\right|\right], \tag{7}$$

is the negative log likelihood associated with the $T$ pseudo datasets and $\alpha_1, \alpha_2 > 0$ controls the strength of regularisation. Note that in comparison with Eqn 4, the latent distributions in $L'$ are informed by historical samples $\{\hat{\mathbf{z}}_{1:N_t,t}\}$ as we expect $f_\theta$ to be able to map $X'_{j,t}$ back to latent variables that resemble samples drawn from $p(\cdot|\hat{\mathbf{z}}_{1:N_t,t})$. See also Fig 1 (b).

**Class-incremental learning**

So far we focused on task-incremental learning where all samples associated with a new task are presented to the model as a single dataset $\mathcal{D}_{T+1}$. It is not always the case in practice as samples associated with the same task may also come in batches. In particular, each batch may contain labels not seen in any previous batches. This scenario is known as *Class-Incremental learning* (CIL) (Zhou et al., 2024). We here demonstrate how CL-BRUNO handles CIL under the scenario where labels in different data batches are disjoint (the extension to the non-disjoint case is straightforward).

Our goal now is to learn from a new batch of data $\mathcal{D}_k^{(1)} = \{X_{i,k}^{(1)}, Y_{i,k}^{(1)}\}_{i=1}^{N_k^{(1)}}$ associated with a *previously seen and known* task $k \in \{1, .., T\}$ such that $Y_{i,k}^{(1)} \neq Y_{j,k}$ for all $Y_{i,k}^{(1)}$ in $\mathcal{D}_k^{(1)}$ and $Y_{j,k}$ in $\mathcal{D}_k$ (i.e. disjoint labels). Such problem can be addressed by updating the normalising flow $f_\theta$ using the same regularisation approach as in Sec 3.1. Similar to $L_{\text{TIL}}$ in Eqn 6, CL-BRUNO under the CIL scenario is updated by finding a $f_\theta$ that minimizes a regularised augmented negative log likelihood

$$L_{\text{CIL}}(\theta) = \overbrace{-\log p_{\theta, \hat{\lambda}_k^{(T)}}(\mathcal{X}_k^{(1)}|k, \mathcal{Y}_k^{(1)}, \mathcal{D}_k)}^{\text{neg log lkd of new batch given old}} + \alpha_1 L'(\theta, \{\hat{\lambda}_t^{(T)}\}_{t=1}^T; \{\mathcal{D}'_t\}_{t=1}^T) + \alpha_2 R(\theta; \hat{\theta}^{(T)}, \{\mathcal{D}'_t\}_{t=1}^T), \tag{8}$$

where

$$\log p_{\theta, \hat{\lambda}_k^{(T)}}(\mathcal{X}_k^{(1)}|k, \mathcal{Y}_k^{(1)}, \mathcal{D}_k) = \sum_{i=1}^{N_k^{(1)}} \log\left(p_{\hat{\lambda}_k^{(T)}}(\mathbf{z}_{i,k}^{(1)}|\mathbf{z}_{1:i-1,k}^{(1)}, \hat{\mathbf{z}}_{1:N_k,k}) \times \left|\det \frac{\partial f_\theta(X_{i,k}^{(1)}; k, Y_{i,k}^{(1)})}{\partial X_{i,k}^{(1)}}\right|\right), \tag{9}$$

is the conditional likelihood of the new feature set $\mathcal{X}_k^{(1)} = \{X_{i,k}^{(1)}\}_{i=1}^{N_k^{(1)}}$ given the corresponding label set $\mathcal{Y}_k^{(1)} = \{Y_{i,k}^{(1)}\}_{i=1}^{N_k^{(1)}}$ and previous data batch $\mathcal{D}_k$ associated with the same task, and $\mathbf{z}_{i,k}^{(1)} = f_\theta(X_{i,k}^{(1)}; k, Y_{i,k}^{(1)})$ is the transformed latent variable generated by the new normalising flow $f_\theta$. Note that given the predictive latent distribution $p_{\hat{\lambda}_k^{(T)}}(\cdot | \hat{\mathbf{z}}_{1:N_k,k})$ from the old CL-BRUNO, $p_{\hat{\lambda}_k^{(T)}}(\mathbf{z}_{i,k}^{(1)} | \mathbf{z}_{1:i-1,k}^{(1)}, \hat{\mathbf{z}}_{1:N_k,k})$ in 9 can be evaluated efficiently using the recursive formula in Korshunova et al. (2020) for all $i = 1, \dots, N_k^{(1)}$ without access to $\hat{\mathbf{z}}_{1:N_k,k}$. In comparison with 6, the parameter of latent distribution $\hat{\lambda}_k^{(T)}$ associated with task $k$ in $L_{\mathrm{CIL}}$ is taken from the old CL-BRUNO and *not updated* alongside with $\theta$. We choose to do so as it simplifies the computation and ensures the tractability of $p_{\hat{\lambda}_k^{(T)}}(\cdot | \hat{\mathbf{z}}_{1:N_k,k})$. This choice can also be viewed as an additional regularisation that aims to mitigate the impact of catastrophic forgetting. See Supplementary material A.1 for additional discussion on $L_{\mathrm{CIL}}$.

## 3.2 Label and task identity prediction

In the previous section, we discussed a unified incremental learning strategy for both task- and class-incremental learning. In particular, we focused on incrementally modelling the feature vectors $p(\mathcal{X}_t | t, \mathcal{Y}_t)$ for tasks $t = 1, \dots, T$. In this section, we discuss the probabilistic inference pipeline for label and task identity prediction. Let $f_{\hat{\theta}^{(T)}}$, $\{p_{\hat{\lambda}_t^{(T)}}(\cdot | \hat{\mathbf{z}}_{1:N_t,t})\}_{t=1}^T$ be the normalising flow and predictive latent distributions trained over $T$ tasks. Let $X^*$ be a generic test point and $Y^*$ be the unknown label of interest. We start from the case where the task identity $t$ is known. Under the assumption given in 2, we approximate the posterior label distribution $p(Y^* | t, X^*, \mathcal{D}_t)$ by

$$\hat{p}(Y^* | t, X^*, \mathcal{D}_t) \propto p_{\hat{\theta}^{(T)}, \hat{\lambda}_t}(X^* | t, Y^*, \mathcal{D}_t) p(Y^* | t) \propto p_{\hat{\lambda}_t}(\mathbf{z}_{Y^*,t}^* | \hat{\mathbf{z}}_{1:N_t,t}) \left| \det \frac{\partial f_{\hat{\theta}^{(T)}}(X^*; t, Y^*)}{\partial X^*} \right| p(Y^* | t), \quad (10)$$

for all $Y^* \in \{1, \dots, C_t\}$, where $\mathbf{z}_{Y^*,t}^* = f_{\hat{\theta}^{(T)}}(X^*; t, Y^*)$ is the transformed latent variable associated with $X^*$ conditioned on label $Y^*$, and $p(Y^* | t)$ is the prior distribution over the $C_t$ classes associated with the $t$-th task.

However, the task identity associated with $X^*$ in practice is not always available, and task identity estimation is itself a challenging problem in CL (Lee et al., 2020). Here we give an easy-to-interpret probabilistic estimate of task identity of a test point $X^*$ under the assumption that $X^*$ is indeed a sample from the underlying generative process of one of the $T$ previously seen datasets $\{\mathcal{D}_t\}_{t=1}^T$ (See Supplementary material A.2 for discussion on other possibilities). Denote $p(t)$ a user specified prior over the $T$ tasks, and $\mathcal{C}_t = \{1, \dots, C_t\}$ the label set associated with the $t$-th task. We approximate the task identity distribution $p(X^*$ from task $t | \{\mathcal{D}_t\}_{t=1}^T)$ for $t = 1, \dots, T$ by

$$\hat{p}(X^* \text{ from task } t | \{\mathcal{D}_t\}_{t=1}^T) \propto p(t) \sum_{Y^* \in \mathcal{C}_t} p_{\hat{\theta}^{(T)}, \hat{\lambda}_t}(X^* | t, Y^*, \mathcal{D}_t) p(Y^* | t), \quad (11)$$

which can be interpreted as the predictive likelihood of observing $X^*$ as the next new sample given previous ones $\mathcal{D}_t$ averaged over all possible labels $Y^*$. We summarise the training and inference procedures of CL-BRUNO for TIL in Alg 1. CL-BRUNO for CIL and their combination work similarly. In addition, if all tasks share the same label space, we can then further marginalise out the uncertainty of task identity in label prediction by

$$\hat{p}(Y^* | X^*, \{\mathcal{D}_t\}_{t=1}^T) = \sum_{t=1}^T \hat{p}(Y^* | t, X^*, \mathcal{D}_t) \times \hat{p}(X^* \text{ from task } t | \{\mathcal{D}_t\}_{t=1}^T). \quad (12)$$

This is particularly useful in biomedical settings where a single CL model incrementally learns to distinguish healthy vs unhealthy from a sequence of datasets collected from patients from different hospitals or regions (i.e. different tasks).

---

**Algorithm 1:** CL-BRUNO for TIL

---

**Data:** Dataset $\{\mathcal{D}_t\}_{t=1}^T$ of the initial $T$ tasks; Regularisation parameters $\alpha_1, \alpha_2 \geq 0$; Pseudo data size $N' \in \mathbb{N}^+$; Datasets $\{\mathcal{D}_{T+t'}\}_{t'=1}^{T'}$ of the subsequent $(T+1)$th to $(T+T')$th tasks.
**Result:** Predicted label and task identity of a generic input $X^*$.

Train the initial model from scratch by finding $\{\hat{\theta}^{(T)}, \hat{\boldsymbol{\lambda}}\}$ minimising $L(\theta, \boldsymbol{\lambda}; \{\mathcal{D}_t\}_{t=1}^T)$ in Eq 4 using stochastic gradient descent.
**for** $t' \leftarrow 1$ **to** $T'$ **do**
    /* Here we assume that the model incorporates one new task $\mathcal{D}_{T+t'}$ at a time.
       Incorporating multiple tasks at the same time works similarly.      */
    Update the initial model by finding $\{\hat{\theta}^{(T+t')}, \hat{\lambda}_{T+t'}\}$ minimising $L_{TIL}(\theta, \lambda_{T+t'})$ given by Eq 6, which depends on the regularisation hyperparameters $\alpha_1, \alpha_2, N'$, using stochastic gradient descent.
**end**

**return** *The predicted task identity distribution $\hat{p}(X^* \text{ from task } t | \{\mathcal{D}_t\}_{t=1}^{T+T'})$ in Eq 11 and label distributions $\hat{p}(Y^*|t, X^*, \mathcal{D}_t)$ in Eq 10 for all $t = 1, ..., (T+T')$ of some generic data $X^*$.*

---

# 4 Related works

## 4.1 Bayesian continual learning

Various Bayesian continual learning methods have been developed for different tasks (Adel, 2025): Bonnet et al. (2025) and Yan et al. (2022) use sequential Bayesian inference to update parameters of Bayesian neural networks while retaining the knowledge learnt from historical data. Raichur et al. (2025) combines Bayesian learning-driven dynamic weighting mechanism (Kendall et al., 2018) and contrastive learning to continually update both the parameters and the architectures of a neural network model in a CIL setting. Jha et al. (2024) propose Neural Process Continual Learning (NPCL), which combines attentive Neural Processes (Kim et al., 2019) and experience replay (Chaudhry et al., 2019), under a Bayesian updating framework. Both CL-BRUNO and NPCL use Neural processes to capture uncertainty in prediction. However, we would like to highlight that NPCL uses Neural Processes to model the posterior distributions of labels $p(Y^*|t, X^*, \mathcal{D}_{prev})$ as a random function of task identity $t$, test point $X^*$ and historical data $\mathcal{D}_{prev}$. In contrast, our approach uses Neural Processes to model the distribution of feature vectors $p(\mathcal{X}_t|t, \mathcal{Y}_t)$ associated with different tasks and labels. Unlike NPCL, our method leverages generative replay to retain learned knowledge without the need for storing any previous samples, making it more appealing to applications where data privacy or storage is of concern. In addition, thanks to the specific covariance function used in C-BRUNO, our proposed method scales linearly with both the sample size and the dimension of feature vector. In contrast, NPCL exhibits quadratic complexity relative to the training sample size due to the attention architecture.

## 4.2 Conditional generative continual learning

The generative replay strategy used in our proposed method is closely related to Continual Learning for Conditional Generation (CLCG) (Wang et al., 2024), which also aims to mitigate catastrophic forgetting by recovering previously-learned data distributions. Recent CLCG methods such as MeRGANs Wu et al. (2018), Boo-VAE (Egorov et al., 2021), Hyper-LifelongGAN (Zhai et al., 2021) and FILIT (Chen et al., 2022) use GAN (Goodfellow et al., 2020) or VAE (Kingma and Welling, 2013) as their underlying conditional generative models and require separate discriminative models for label prediction. Scardapane et al. (2020) uses normalizing flows to retain distributional information at the feature embedding level, but still requires separate encoders and discriminators. In comparison, CL-BRUNO utilises a tractable deep generative model that supports both conditional generation and density estimation on feature vectors. This enables tractable, efficient, and statistically principled label prediction and task identity estimation without the need for separate discriminators or any other modules, improving both interpretability and computational efficiency.

## 5 Experiments

In this section, we first empirically evaluate our algorithm on four image benchmarking datasets. We then demonstrate the efficacy and versatility of CL-BRUNO using two real-world biomedical datasets.

### 5.1 Benchmarking on image classification tasks

We evaluate the proposed method on both class and task incremental learning (IL) settings. For class-IL (CIL), we use three public datasets: incremental CIFAR10 (iCIFAR-10) (Lopez-Paz and Ranzato, 2017), incremental CIFAR100(iCIFAR-100) (Zenke et al., 2017), and sequential Tiny ImageNet (S-TinyImageNet) (Chaudhry et al., 2019). For task-IL(TIL), we use the MNIST dataset (LeCun, 1998). iCIFAR-10, iCIFAR100, and S-TinyImageNet contain 3-channel images of size $32 \times 32$, $32 \times 32$ and $64 \times 64$ from 10, 100 and 200 classes, and each class includes 5000, 500, 500 training images and 500, 50, 50 test images. The MNIST dataset consists of single-channel $28 \times 28$ images of handwritten digits from 0 to 9, and each digit includes $\sim 6000$ training images and $\sim 1000$ test images. Under the CIL setting, the numbers of sequential batches for iCIFAR-10, iCIFAR-100 and S-Tiny-ImageNet are 5 (2 classes per batch), 10 (10 classes per batch) and 10 (20 classes per batch) respectively. Under the TIL setting, we follow the experiment setup in Egorov et al. (2021), and split the MNIST dataset into 5 batches, where each batch consists of images of two digits. We view each batch as a binary classification task (0 vs 1, 2 vs 3, ..., 8 vs 9).

We compare CL-BRUNO with both experience replay-based methods including ER (Riemer et al., 2018), DER (Buzzega et al., 2020), NPCL (Jha et al., 2024) and CSReL-LODE (Tong et al., 2025), and exemplar-free methods including EWC (Kirkpatrick et al., 2017), oEWC-MACL (Wang and Huang, 2024), LwF (Li and Hoiem, 2017), SSRE (Zhu et al., 2022), DS-AL (Zhuang et al., 2024), MeRGAN (Wu et al., 2018), FeTRIL (Petit et al., 2023), NICE (Gurbuz et al., 2024), PRER (Scardapane et al., 2020), ReReLRP (Bogacka et al., 2025) and Boo-VAE (Egorov et al., 2021). The default or recommended settings are used for all the above methods. For all experience replay-based methods, we fix the buffer size $M = 500$.

Our CL-BRUNO is specified as follows: We use a Resnet18 (He et al., 2016) as a feature extractor (i.e. using the output of the second-last layer of Resnet18, a 512-dimensional real vector, as the transformed feature of the corresponding input image) and apply CL-BRUNO to the 512-dimensional feature vectors. Specifically, the Resnet18 feature extractor here is only trained using the images from the first batch of data in the initial state, and is then frozen for the reminder of the class- or task-incremental learning process. Similar strategies have been used in exemplar-free methods such as Petit et al. (2023). (In principle, we could directly specify a generative model on raw images instead of feature vectors. However, we do not consider it here as our method is motivated by tabular rather than image data.) In this example, we set the number of coupling layers in CL-BRUNO to be 6, the dimension of class embedding to be 128, size of pseudo data $N' = 128$ (i.e. $N'$ pseudo samples are generated to compute the distributional regulariser in Eq 6 or 8 for every gradient descent step) and regularisation strength $\alpha_1 = \alpha_2 = 1$. We report the classification accuracy on test set for each of the methods in Table 1. We see that CL-BRUNO either outperforms existing exemplar-free methods or achieves competitive performance across multiple experimental settings. We further report the ranking of the performance of different methods in Supp Fig 4. Note that CL-BRUNO does not perform as well on S-TinyImagenet. This is likely due to the fact that the fixed feature extractor trained on the first batch may not be sufficient to capture the distributional difference in the forthcoming data.

#### Memory overhead

We then investigate the memory overhead of CL-BRUNO in comparison with other methods using the iCIFAR100 dataset as an example. Since we follow the setup in Jha et al. (2024) and Boschini et al. (2022), CL-BRUNO use Resnet18, which consists of 11.7M parameters, as a backbone model. In addition to the backbone model, CL-BRUNO also maintains a C-BRUNO model consisting of $\sim 1.38$M parameters, and 100 class embeddings (12.8K parameters). As a result, CL-BRUNO in total requires storing $\sim 1.39$M extra parameters in addition to the backbone Resnet18 model. In comparison, the replay-based method requires storing exemplar samples from historical datasets instead of extra model parameters. In our setup where the buffer size is fixed at 500, maintaining this buffer requires storing $\sim 1.54$M floating point numbers, which

| Type | Method | iCIFAR10 CIL | iCIFAR100 CIL | S-TinyImagenet CIL | MNIST TIL |
|---|---|---|---|---|---|
| (a) | ER | 0.577 | 0.221 | 0.099 | 0.972 |
| | DER | 0.705 | 0.366 | 0.178 | 0.983 |
| | NPCL[3] | 0.702 | 0.198 | 0.135 | - |
| | CSReL-LODE | 0.398 | 0.420 | 0.228 | - |
| (b) | EWC | 0.182 | 0.092 | 0.056 | 0.441 |
| | oEWC-MACL | 0.206 | 0.088 | 0.079 | - |
| | LwF | 0.196 | 0.145 | 0.094 | 0.472 |
| | SSRE | 0.355 | 0.179 | 0.110 | - |
| | DS-AL | 0.408 | 0.205 | **0.122** | - |
| | MeRGAN | 0.251 | 0.122 | 0.078 | 0.670 |
| | FeTriL | 0.417 | 0.208 | 0.115 | - |
| | NICE | **0.551** | 0.203 | 0.120 | - |
| | PRER | 0.412 | 0.197 | 0.108 | - |
| | ReReLRP | 0.399 | 0.058 | 0.033 | - |
| | Boo-VAE | - | - | - | 0.892 |
| | CL-BRUNO$_{\alpha_2=0}$ | 0.257 | 0.146 | 0.056 | 0.734 |
| | CL-BRUNO | 0.421 | **0.212** | 0.117 | **0.947** |

Table 1: Classification accuracy of different CL methods on iCIFAR10, iCIFAR100, S-TinyImagenet and MNIST. - indicates method is not applicable to the problem setting. (a) Experience replay with buffer size $M = 500$ (b) Exemplar-free. The best and second best results in (b) are highlighted in **boldface** and underline respectively. CL-BRUNO$_{\alpha_2=0}$ refers to CL-BRUNO without functional regulariser.

is larger than the memory required by CL-BRUNO. In addition to exemplar samples, NPCL also requires maintaining an attention-based network consisting of another $\sim$12.8M parameters. Among exemplar-free models, EWC, LwF, oEWC-MACL and DS-AL maintain single discriminative models based on Resnet18 or Resnet34, which consists of $\sim 11.7$M and $\sim 21.1$M parameters respectively. MeRGAN requires $\sim$7.15M parameters under the default setting. FeTriL has a smaller extra memory requirement ($\sim$0.3M parameters) in addition to the backbone Resnet18 or Resnet34 as it encodes and stores historical data sets as fixed-length vectors instead of a full generative model. Although MeRGAN and FeTriL require less memory than CL-BRUNO, we see that CL-BRUNO outperforms them in terms of classification accuracy. In addition, we would like to highlight that CL-BRUNO is more versatile than exemplar-free CIL methods as it offers additional functionalities such as task incremental learning (Sec 3.1), task identity estimation (Sec 3.2) and outlier detection (Supplementary material A.2).

**Prediction calibration**

To demonstrate CL-BRUNO's uncertainty quantification feature, we compare the Expected Calibration Error (ECE) (Guo et al., 2017) of CL-BRUNO with both experience-replay and exemplar-free CL baselines in a similar fashion to (Jha et al., 2024). In Table 2 we see CL-BRUNO attains lower ECE than exemplar-free CL methods with similar accuracies (FeTriL and NICE), and show comparable performance to experience-replay methods on both iCIFAR10 and iCIFAR100 datasets. This suggests that the probabilistic nature of CL-BRUNO benefits it in terms of prediction calibration.

---

[3]Jha et al. (2024) reported classification accuracy 37.4% and 15.3% on the iCIFAR100 and S-TinyImagenet dataset. Our independent run using the publicly available implementation provided by the authors under the same setup gave classification accuracies of 19.8% and 13.5% respectively. Our code for reproducing the NPCL results on iCIFAR100 and S-TinyImagenet can be found here.

| Dataset/Method | Experience replay | | | Exemplar-free | | |
| --- | --- | --- | --- | --- | --- | --- |
| | ER | DER | NPCL | FeTriL | NICE | CL-BRUNO |
| iCIFAR10 | 0.455 | 0.299 | 0.210 | 0.310 | 0.268 | **0.245** |
| iCIFAR100 | 0.646 | 0.248 | 0.199 | 0.358 | 0.472 | **0.303** |

Table 2: Expected Calibration Error (ECE) of different methods. Lowest ECE in Exemplar-free methods is highlighted in **boldface**.

**Ablation study**

In Sec 3 we have demonstrated how the distributional regulariser $L'$ naturally arises from the Bayesian updating rule. However, the inclusion of the functional regulariser $R$ is not justified. Therefore, we first investigate the role of the functional regulariser $R$ in the training of CL-BRUNO. We apply CL-BRUNO without the functional regulariser (CL-BRUNO$_{\alpha_2=0}$) to the same datasets described above, and report the classification accuracies in Table 1. We see that the default CL-BRUNO outperforms CL-BRUNO$_{\alpha_2=0}$ by a large margin across all experimental settings. This confirms that the functional regulariser $R$ contributes to the performance of CL-BRUNO.

We next investigate the size of pseudo data $N'$. We use the iCIFAR100 dataset and run CL-BRUNO under four choices of pseudo samples $N'_1 = 32, N'_2 = 64, N'_3 = 128$ and $N'_4 = 256$. The final classification accuracy of the four models is 10.9%, 17.1%, 21.2% and 22.8%, respectively, showing that increasing $N'$ leads to a diminishing increase in classification accuracy. We further investigate the impact of $N'$ on the computational cost of the proposed method. The running times of the four models are 2569s, 2726s, 2917s, and 3562s, respectively. We see that increasing $N'$ by a factor of 8 (from 32 to 256) leads to a $\sim 40\%$ increase in the running time of CL-BRUNO. This also confirms that increasing $N'$ does not drastically increase the computational cost of CL-BRUNO.

## 5.2 Pan-Cancer Atlas dataset

We here demonstrate CL-BRUNO under a CIL scenario using the Pan-Cancer Atlas (PANCAN) dataset (Hoadley et al., 2018). The PANCAN dataset consists of pre-processed RNAseq readings of $N = 10,535$ tumour samples from 33 cancer types. In our analysis, we use only the top $P = 2,000$ most variable genes as the feature vector associated with each tumour sample, and split the PANCAN dataset into 6 groups according to their cancer types: the first consists of 8 cancer types, while each of the rest consists of 5 cancer types. Cancer types are partitioned in a way such that all 6 groups of data have similar sample sizes. Our goal is to predict cancer type from the RNAseq data of a tumour under a CIL scenario, where 6 groups of data are presented to the model sequentially. In this example, we set the number of coupling layers in CL-BRUNO to be 6, the dimension of task and label embedding to be 256, size of pseudo data $N' = 128$ (i.e. $N'$ pseudo samples are generated to compute the distributional regulariser in Eq 6 or 8 for every gradient descent step) and regularisation strength $\alpha_1 = \alpha_2 = 1$. Each group of data is randomly split into a training set consisting of 80% of the samples and a test set containing the rest, and the performance is measured by the misclassification rate on all test sets after the model has been incrementally trained on all training sets. We compare CL-BRUNO with three exemplar-free CL methods: EWC (Kirkpatrick et al., 2017), LwF (Li and Hoiem, 2017) and MeR-GAN (Wu et al., 2018) under default or recommended settings. The misclassification rates on test sets are reported in Table 3. We also demonstrate in Fig 2(a) how CL-BRUNO retains previously learnt knowledge by reporting the misclassification rate associated with each of the 6 test sets at each incremental learning step, and compare them with results from an oracle CL-BRUNO model that has access to all historical data (i.e. trained by directly minimising Eqn 4) at each incremental learning step. Figure 2 (a) shows how CL-BRUNO retains historical knowledge in terms of prediction accuracy compared to the oracle model.

## 5.3 Immune Checkpoint Inhibitors dataset

We also tested CL-BRUNO under a TIL scenario using the Molecular Response to Immune Checkpoint Inhibitors (ICI) dataset (Eddy et al., 2020). Immune Checkpoint Inhibitors are designed to block (checkpoint)

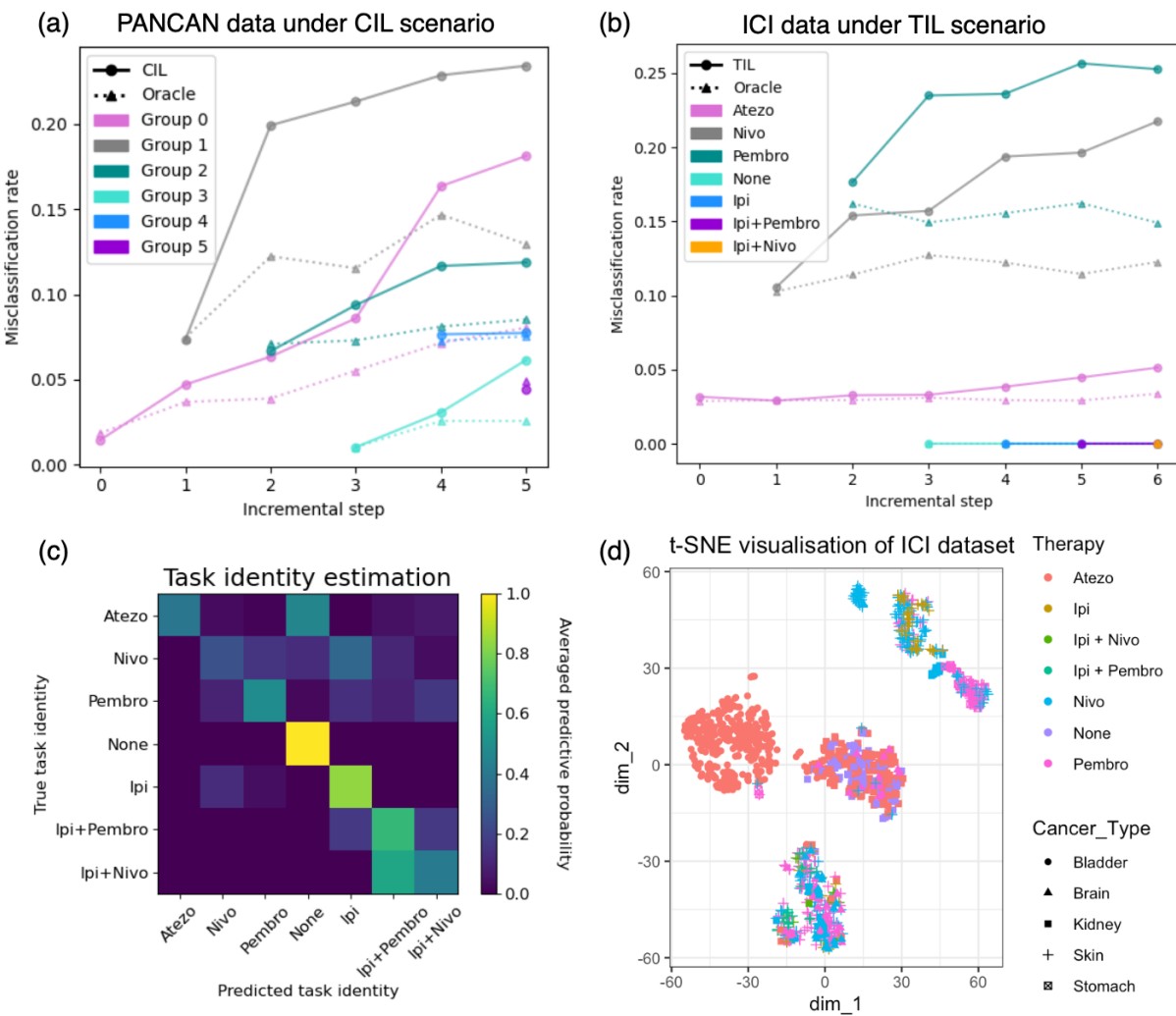

Figure 2: **(a-b) Evolution of misclassification rate specific to each incremental PANCAN batch.** Each point represents the misclassification rate specific to an incremental dataset evaluated at a specific training step using the incrementally trained CL-BRUNO. Each triangle represents the same quantity given by a CL-BRUNO who has access to all historical datasets (oracle). (a) PANCAN dataset under a CIL scenario, (b) ICI dataset under a TIL scenario. Note that ICI dataset contains tasks with only one class, which leads to zero test error. **(c)-(d) ICI dataset**. (c): Heat map of predicted task identity. Each row corresponds to the categorical task identity distribution 11 averaged over test samples from each task. (d): t-SNE (Van der Maaten and Hinton, 2008) projection of the pre-processed RNAseq measurement associated with different therapy types in ICI dataset.

proteins expressed by some cancers which allow it to avoid detection by the immune system. Blocking the checkpoints, reactivates the immune system to target the cancer. The ICI dataset contains pre-processed RNAseq data from $N = 1,142$ patients' tumour samples under 7 different types of immunotherapy treatments. This included four single therapy only regimes: Atezolizumab (`Atezo`), Nivolumab (`Nivo`), Pembrolizumab (`Pembro`), Ipilimumab (`Ipi`), two combination therapies (`Ipi+Pembro`, `Ipi+Nivo`) and a non-treatment/placebo group (`None`). Each ICI is designed to block a certain checkpoint protein. As a consequence, ICIs are only effective against those cancers that express the targeted checkpoint protein. Our goal is to determine if CL-BRUNO could learn the molecular signatures associated with immunotherapy effectiveness which is by proxy a cancer type prediction problem given the RNAseq data. In this example, we split the dataset into 7 groups

| Method/Data | PANCAN | ICI |
|---|---|---|
| CL-BRUNO | **0.139**(**0.0233**) | **0.118**(**0.0351**) |
| EWC | 0.569(0.0265) | 0.331(0.0392) |
| LwF | 0.322(0.0217) | 0.157 (0.0433) |
| MeR-GAN | 0.518(0.0521) | - |

Table 3: **Misclassification rate of different methods on PANCAN and ICI datasets**. Results are averaged over 5 repeated runs. Standard deviations of the misclassification rates are reported in brackets.- indicates method is not applicable to the problem setting. Best results are in **boldface**.

according to the therapy the patients received, and treat the classification problem under each therapy as an individual task. The 7 tasks consists of $N_1 = 524, N_2 = 224, N_3 = 218, N_4 = 89, N_5 = 42, N_6 = 32, N_7 = 13$ samples and $C_1 = 2$ ({Bladder, Kidney}), $C_2 = 3$ ({Brain, Skin, Kidney}), $C_3 = 3$ ({Stomach, Brain, Skin}), $C_4 = 1$ ({Kidney}), $C_5 = 1$ ({Skin}), $C_6 = 1$ ({Skin}), $C_7 = 2$ ({Kidney, Skin}) unique cancer types respectively. We use the same set of hyperparameters and training strategy as in the last example to train the CL-BRUNO, and compare its performance with EWC and LwF under default or recommended settings.[4] This example is challenging due to the relatively small and imbalanced sample sizes. We report the misclassification rates given by different methods in Table 3. The knowledge retention curves compared to the oracle model is reported in Fig 2(b). We report the task identity estimates given by the trained CL-BRUNO model under a uniform task prior in Fig 2(c). We see that the task identity estimates put most of the probability mass on the correct task identities (the diagonal line), indicating good prediction accuracy.

In addition to prediction accuracy, we also demonstrate that CL-BRUNO is able to give probabilistic predictions that correctly reflects the uncertainty when tasks are indeed indistinguishable: Pal et al. (2022) report that `Atezo` is ineffective for kidney cancers. Hence we expect patients given `Atezo` for kidney cancer to be indistinguishable from patients who received no ICI treatment `None` as both are effectively equivalent to no treatment. To verify if CL-BRUNO captures this relationship, we first split the test set of `Atezo` into 4 subgroups depending on their cancer type (Kidney vs Non-kidney) and response to the therapy (Responder vs Non-responder), then compute the averaged predicted probabilities separately. From Fig 3(c) we see CL-BRUNO is much more likely to assign kidney cancer patients given `Atezo` to `None` in comparison with non-kidney cancer patients regardless of the response status. This agrees with previous studies, and is also confirmed by visualisation, as we see from Fig 3 (a) that patients from `None` overlap with a cluster of kidney cancer samples in `Atezo`.

By the same rationale, since both `Nivo` and `Ipi` are given to patients with skin cancer, we expect non-responders in `Nivo` with skin cancer to be indistinguishable from non-responders in `Ipi`, which consists of solely patients with skin cancer. To verify if CL-BRUNO captures this relationship, we split the test set of `Nivo` and compute the averaged predicted probabilities in a similar fashion as before. From Fig 3 (d) we see non-responders in `Nivo` with skin cancer are much more likely to be classified as `Ipi` in comparison with the rest. This is supported by the visualisation in Fig 3 (b) as we can see a clear overlap between the non-responders to `Nivo` and `Ipi` with skin cancer. This unique pattern suggests that CL-BRUNO accurately captures the inter-task relationship between `Nivo` and `Ipi`, and further confirms that CL-BRUNO is capable of capturing inter-task relationships under a TIL scenario.

# 6 Conclusion

We propose CL-BRUNO, a probabilistic, exemplar-free CL model based on exchangeable sequence modelling. Compared with existing generative continual learning methods, our proposed method provides a unified probabilistic framework capable of handling different types of CL problems such as TIL and CIL, and giving easy-to-interpret probabilistic predictions without the need of training or maintaining a separate classifier. These features make our approach appealing in applications where data privacy and uncertainty quantification are of concern.

---

[4]We did not include MeR-GAN here as it is designed for CIL.

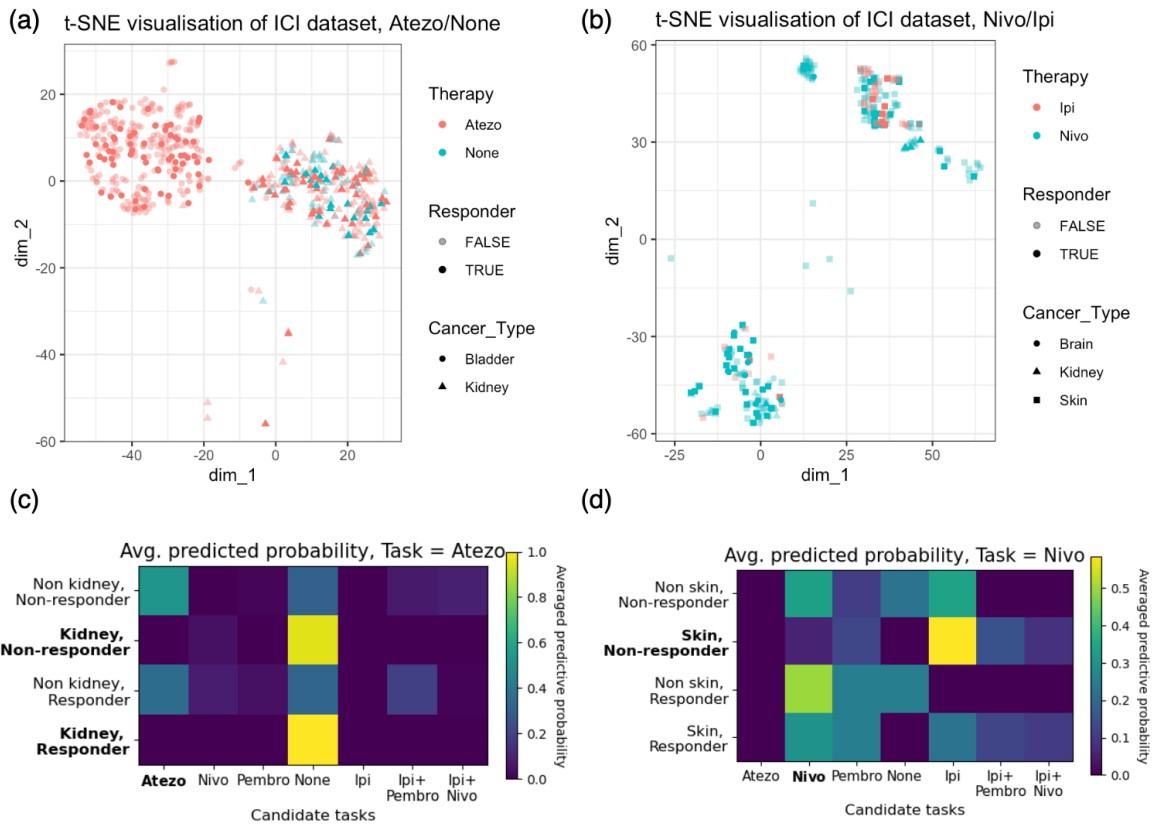

Figure 3: **Visualisation of subsets of ICI dataset** (a): t-SNE projection of samples from `Atezo` and `None`. (b): t-SNE projection of samples from `Nivo` and `Ipi`. (c): Averaged predicted probabilities for different groups of patients under treatment `Atezo`. Patients are split into four groups based on cancer type (Kidney cancer vs Non-kidney cancer) and responsiveness to treatment (Responder vs Non-responder). (d): Averaged predicted probabilities for groups of patients under treatment `Nivo`. Patients are split into four groups in a similar fashion to (c).

## Broader Impact Statement

In this work, we propose CL-BRUNO, a generative-replay approach for continual learning that avoids storing explicit exemplar samples, thereby offering a potential advantage in privacy-sensitive applications. We acknowledge that generative models trained on real data can encode and reproduce distributional and/or identifiable patterns of the training data. One potential solution to this issue is to incorporate pretrained feature extractors in a similar fashion to the image datasets. By doing so, the model would only remember the distributional patterns of the feature vectors instead of the original data, hence preventing the model from reproducing the original training data. Although our experiments do not involve user-specific or sensitive biometric data (PANCAN and ICI datasets are both non-identifiable data sets made available for research), we recognise that applying such models to real-world, privacy-critical domains would necessitate more rigorous safeguards. Future work should incorporate privacy-preserving techniques, e.g. differential privacy, to ensure that generated samples do not leak sensitive information.

We also recognise the potential for bias amplification in continual learning settings, particularly when using biometric datasets that may exhibit class imbalance or uneven demographic representation. Although our current study focuses on methodological contributions rather than fairness-specific evaluations, we believe that integrating fairness-aware strategies is essential for real-world deployment. Addressing these

ethical considerations is a key direction for future research, particularly for applications involving socially or demographically sensitive data.

## Acknowledgements

The authors are supported by an EPSRC Turing AI Acceleration Fellowship (Grant Ref: EP/V023233/1).

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

# A  Additional discussion on CL-BRUNO

## A.1  Discussion on CIL loss

In this section, we justify the choice of $L_{CIL}$ in Sec 3.1. $L_{CIL}$ takes the form

$$L_{\text{CIL}}(\theta) = \overbrace{-\log p_{\theta,\hat{\lambda}_k^{(T)}}(\mathcal{X}_k^{(1)}|k,\mathcal{Y}_k^{(1)},\mathcal{D}_k)}^{\text{neg log lkd of new batch given old}} + \alpha_1 L'(\theta, \{\hat{\lambda}_t^{(T)}\}_{t=1}^T; \{\mathcal{D}_t'\}_{t=1}^T) + \alpha_2 R(\theta; \hat{\theta}^{(T)}, \{\mathcal{D}_t'\}_{t=1}^T), \quad (6)$$

where

$$\log p_{\theta,\hat{\lambda}_k^{(T)}}(\mathcal{X}_k^{(1)}|k,\mathcal{Y}_k^{(1)},\mathcal{D}_k) = \sum_{i=1}^{N_k^{(1)}} \log \left( p_{\hat{\lambda}_k^{(T)}}(\mathbf{z}_{i,k}^{(1)}|\mathbf{z}_{1:i-1,k}^{(1)}, \hat{\mathbf{z}}_{1:N_k,k}) \times \left| \det \frac{\partial f_\theta(X_{i,k}^{(1)};k,Y_{i,k}^{(1)})}{\partial X_{i,k}^{(1)}} \right| \right). \quad (7)$$

Recall that under the TIL scenario, the trained CL-BRUNO does not depend on the ordering of samples in each dataset, as we assumed that samples within each task are exchangeable. Similarly, under the CIL scenario, we also expect the trained model to *not depend* on the ordering of data batches the model has been trained on (e.g. a model first trained on $\{\mathcal{X}_k, \mathcal{Y}_k\}$ and then $\{\mathcal{X}_k^{(1)}, \mathcal{Y}_k^{(1)}\}$ should be the same as a model trained on the reverse order). This requirement is easily satisfied by our CL-BRUNO model under the assumption that the concatenated $\{\mathcal{X}_k, \mathcal{X}_k^{(1)}\}$ is jointly exchangeable given $\{\mathcal{Y}_k, \mathcal{Y}_k^{(1)}\}$, i.e. the two data batches $\{\mathcal{X}_k, \mathcal{Y}_k\}$ $\{\mathcal{X}_k^{(1)}, \mathcal{Y}_k^{(1)}\}$ associated with the $k$-th task are "segments" of a sequence generated from an underlying exchangeable generative process.

In particular, under the distributional assumption above, CL-BRUNO aims to learn the joint distribution $p(\mathcal{X}_k^{(1)}, \mathcal{X}_k|t, \mathcal{Y}_k^{(1)}, \mathcal{Y}_k)$ of the concatenated exchangeable sequence $\{\mathcal{X}_k^{(1)}, \mathcal{X}_k\}$ given the label $\{\mathcal{Y}_k^{(1)}, \mathcal{Y}_k\}$ while retaining the marginal $p_{\hat{\theta}^{(T)}, \hat{\lambda}_k^{(T)}}(\mathcal{X}_k|t, \mathcal{Y}_k)$ the model has learnt from historical dataset. In other words, terms in $L_{CIL}$ associated with the $k$-th task can be interpreted as learning the joint

$$p(\mathcal{X}_k^{(1)}, \mathcal{X}_k|t, \mathcal{Y}_k^{(1)}, \mathcal{Y}_k) = p(\mathcal{X}_k^{(1)}|t, \mathcal{Y}_k^{(1)}, \mathcal{X}_k, \mathcal{Y}_k)p(\mathcal{X}_k|t, \mathcal{Y}_k)$$

under the constraint that

$$p(\mathcal{X}_k|t, \mathcal{Y}_k) = p_{\hat{\theta}^{(T)}, \hat{\lambda}_k^{(T)}}(\mathcal{X}_k|t, \mathcal{Y}_k).$$

Note that the first term in $L_{CIL}$ forces the normalising flow to learn the conditional $p(\mathcal{X}_k^{(1)}|t, \mathcal{Y}_k^{(1)}, \mathcal{X}_k, \mathcal{Y}_k)$ under the marginal constraint, while the regularisers in $L_{CIL}$ related to the $k$-th task in 6 enforce the marginal constraint by penalising both distributional and functional deviation between the new $p_{\theta, \hat{\lambda}_k^{(T)}}(\mathcal{X}_k|t, \mathcal{Y}_k)$ and the old $p_{\hat{\theta}^{(T)}, \hat{\lambda}_k^{(T)}}(\mathcal{X}_k|t, \mathcal{Y}_k)$. The latent distribution parameter $\hat{\lambda}_k^{(T)}$ are not updated alongside with the normalising flow parameter $\theta$. We choose to do so as it simplifies the computation and ensures the tractability of $p_{\hat{\lambda}_k^{(T)}}(\cdot|\hat{\mathbf{z}}_{1:N_k,k})$. This choice can also be viewed as an additional regularisation that aims to mitigate the impact of catastrophic forgetting.

Figure 4: Rankings performance of exemplar-free methods based on their accuracies in Table 1. Note that our proposed CL-BRUNO show competitive performance across all experiments.

## A.2 Outlier detection using CL-BRUNO

In Sec 3.1, we give task-identity estimate of a generic test point $X^*$ under the assumption that $X^*$ is indeed drawn from the generative process of one of the previously seen $\mathcal{D}_t$s. This assumption no longer holds when $X^*$ is e.g. an outlier. Thanks to the tractable generative modelling framework of CL-BRUNO, users can identify outliers in a straightforward fashion using e.g. level sets (Hyndman, 1996): For each task $t$, we first generate a pseudo dataset $\hat{\mathcal{D}}_t$ as in Sec 3.1 and discard the pseudo labels $\hat{Y}_{i,t}$, then for each generated pseudo feature vector $\hat{X}_{i,t}$, $i = 1, ..., N'$, we evaluate

$$\hat{p}_t(\hat{X}_{i,t}) = p_{\hat{\theta}^{(T)}, \hat{\lambda}_t}(\hat{X}_{i,t}|t, \mathcal{D}_t) \tag{8}$$

$$= \sum_{Y^* \in \mathcal{C}_t} p_{\hat{\theta}^{(T)}, \hat{\lambda}_t}(\hat{X}_{i,t}|t, Y^*, \mathcal{D}_t) p(Y^*|t), \tag{9}$$

the approximate predictive density of observing $\hat{X}_{i,t}$ as the next sample conditioned on $\mathcal{D}_t$ marginalised over all labels $Y^* \in \mathcal{C}_t$ where $p(\cdot|t)$ is the prior that generates the pseudo labels $\hat{Y}_{i,t}$. Note that by definition, each $\hat{X}_{i,t}$ is indeed a sample drawn from $p_{\hat{\theta}^{(T)}, \hat{\lambda}_t}(\cdot|t, \mathcal{D}_t)$, the approximate marginal predictive distribution of feature vectors from the $t$-th task learnt by CL-BRUNO under the chosen label prior $p(\cdot|t)$. Let $\alpha \in (0, 1)$ and $\hat{p}_{\alpha,t}$ be the $\alpha\%$ quantile of $\{\hat{p}_t(\hat{X}_{i,t})\}_{i=1}^{N'}$. By Hyndman (1996), level set $S_t = \{X' : \hat{p}_t(X') > \hat{p}_{\alpha,t}\}$ defines a $(1 - \alpha)\%$ approximate highest density region of $p_{\hat{\theta}^{(T)}, \hat{\lambda}_t}(\cdot|t, \mathcal{D}_t)$. As a result, one could test if $X^*$ is a typical feature vector associated with the $t$-th task in a natural and interpretable fashion by comparing $\hat{p}_t(X^*)$ with the threshold $\hat{p}_{\alpha,t}$.

## B   Additional experiments

We include additional synthetic and real-world examples in this section.

### B.1   Synthetic data

Here we demonstrate CL-BRUNO using a synthetic dataset. We set data dimension $D = 1000$, and consider $T = 4$ tasks. Each task $t = 1, ..., T$ is associated with a classification problem with $C_t = t + 1$ distinct classes. Each synthetic dataset $\mathcal{D}_t$ consists of $N_t = 500$ samples, where labels $Y_{i,t} \sim Unif(\{1, ..., C_t\})$, and features $X_{i,t}|Y_{i,t} \sim N(\boldsymbol{\mu}_{i,t}, 0.5\mathbf{I}_D)$ with $\mathbf{I}_D$ being a $D \times D$ identity matrix, $\boldsymbol{\mu}_{i,t} = \{\mu_{i,t}^{(d)}\}_{i=1}^D$ and $\mu_{i,t}^{(d)} = \sqrt{t} \sin(2\pi Y_{i,t}/C_t)$

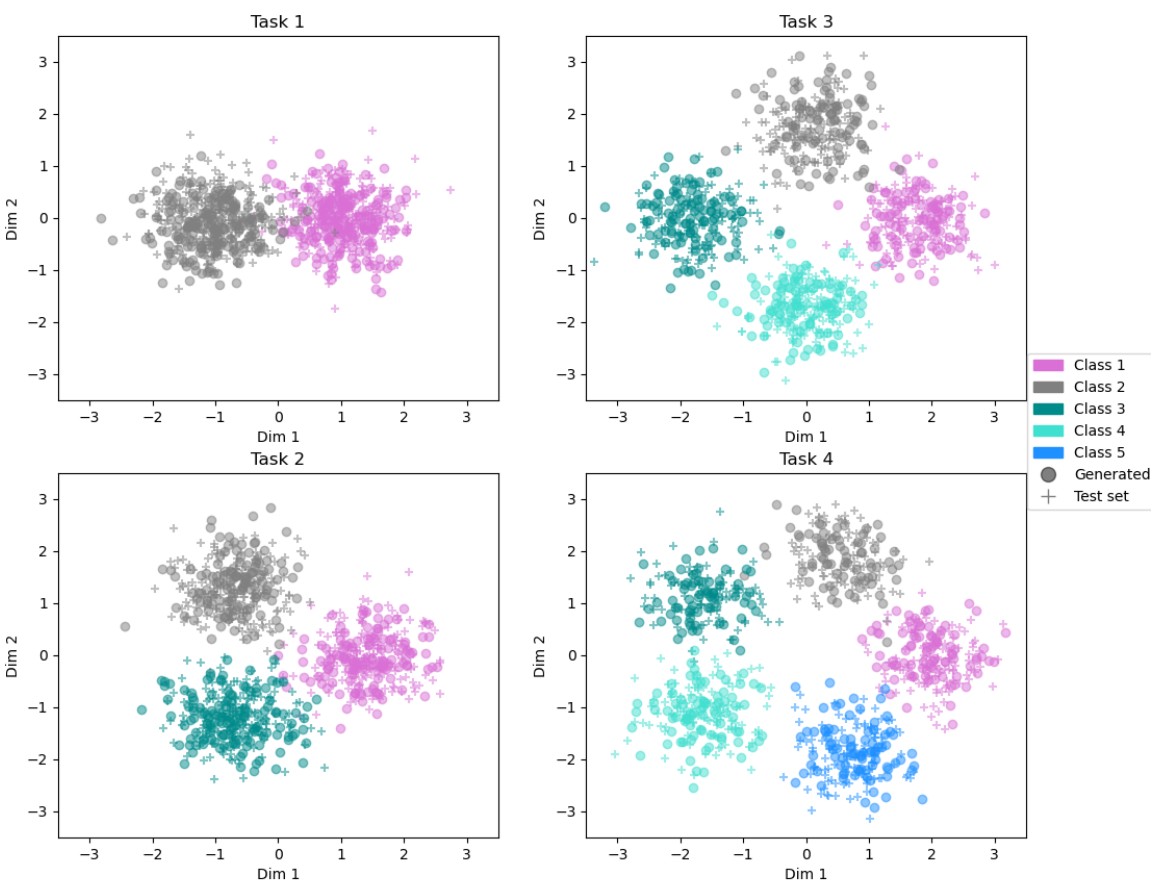

Figure 5: Scatter plots of the first two dimensions of samples in the test set (cross) and samples generated from the trained CL-BRUNO (circle) for each of the 4 tasks.

if $d$ is odd and $\mu_{i,t}^{(d)} = \sqrt{t}\cos(2\pi Y_{i,t}/C_t)$ if $d$ is even. See Fig () for an illustration of the synthetic datasets. We first initialise our proposed model on task 1 by training the generative on $\mathcal{D}_1$, then we incrementally present task 2 ($\mathcal{D}_2$) and 3 ($\mathcal{D}_3$) to the model. We then demonstrate the class-incremental scenario using task 4: We split $\mathcal{D}_4$ into two groups, one containing data associated with the first three out of the five classes in $\mathcal{D}_4$, and the other containing the rest two classes. We then present the two groups of data sequentially to the model. As a result, the example consists of 5 incremental training steps in total (1 for initialisation, 2 for task-incremental learning and 2 for class-incremental learning). We parameterise task ID $t$ and each distinct label associated with task $t$ for each task $t = 1, ..., T$ as trainable embeddings of length 16. After the training phase, we test the classification accuracy on all four tasks using separate test sets drawn from the same generative process. Note that no samples in training data $\{\mathcal{D}\}_{t=1}^{T}$ is retained in our trained model.

For all four tasks, the trained model attains $< 1\%$ misclassification rate on test sets of size 1000 samples. CL-BRUNO is also able to accurately predict task identities of samples from test sets, attaining $< 5\%$ misclassification rate on all tasks. To visualise the fit of CL-BRUNO, we report scatter plots of the first two dimension of samples from the test set and samples generated by the trained CL-BRUNO in Fig 5. We see CL-BRUNO is able to accurately capture the feature distributions associated with each class within each task. This confirms the effectiveness of CL-BRUNO.

## B.2 Discussion on computational cost

We inspect the computational cost of CL-BRUNO in terms of wall clock time using the same S-CIFAR-100 example. All examples are executed on our machine with an AMD Ryzen7 2700 CPU and NVIDIA RTX 2060 GPU. Under the experiment setup described above, CL-BRUNO takes around $2.9 \times 10^3$s to complete. In comparison, the running time of non-generative, exemplar-free LwF and EWC are around $2.4 \times 10^3$s, whereas the generative, exemplar-free FeTriL takes around $3.2 \times 10^3$s. The experience replay-based NPCL takes around $3.5 \times 10^5$s to run. We see in this example that the computational cost of CL-BRUNO is on a scale comparable to the existing exemplar-free CL models.

