# OpenReview forum: "Continual learning via probabilistic exchangeable sequence modelling"
_TMLR — Accepted by TMLR_

### Review · Reviewer_SE8t · 2025-05-21

**Summary Of Contributions:**

The paper proposes CL-BRUNO, a continual learning method based on exchangeable sequence modeling with Neural Processes. The key idea is to treat data observed across tasks as samples from an exchangeable sequence, enabling a unified probabilistic view of continual learning without requiring explicit task identities.

Main contributions include:
* Probabilistic Formulation: The proposed method is modeled as Bayesian inference over exchangeable sequences, enabling task-agnostic continual learning.

* Neural Process-based Implementation: CL-BRUNO uses Neural Processes to amortize inference, achieving scalability and flexibility across tasks.

* No Replay or Task IDs: The method avoids storing past data or relying on task boundaries, relying instead on functional and distributional regularization to mitigate forgetting.

* Empirical Results: Experiments on vision and biomedical datasets show that CL-BRUNO achieves good performance compared to baselines across task-incremental and class-incremental CL settings.

**Audience:**

Yes

**Broader Impact Concerns:**

The submission does not include a Broader Impact Statement, and currently lacks a discussion on the potential ethical implications of deploying continual learning systems in real-world settings. While the proposed framework is primarily evaluated in academic benchmarks, continual learning methods—particularly those designed for biometric data or long-term user modeling—can raise important privacy, fairness, and misuse concerns.

In particular:
* **Privacy risks**: Although the proposed method avoids storing explicit memory samples, it relies on generating synthetic samples for replay. It remains unclear whether these generated samples are truly anonymized and free from privacy leakage. Can such samples inadvertently preserve or reconstruct identifiable information about users? A discussion on the safety and privacy guarantees of the generative module is warranted.

* **Bias amplification**: Continual learning performance is often sensitive to task ordering and task difficulty. This issue is especially pronounced in biometric datasets, which often exhibit class imbalance or uneven data distributions. The manuscript does not address how such biases may affect model behavior or what strategies (e.g., curriculum learning, re-weighting) could be employed to mitigate them. This is important for ensuring fairness across different user groups or identity classes.

I recommend that the authors add a Broader Impact Statement to reflect on these issues, even if only to clarify the intended scope and limitations of their method under ethical considerations.

**Claims And Evidence:**

Yes

**Requested Changes:**

1. I would suggest shifting the focus from your biometric datasets to widely-used benchmarks in the continual learning literature. The current datasets are quite unfamiliar to other researchers and difficult to evaluate. For example, consider adding results on datasets such as ImageNet-200, CORe50, or iCIFAR-100 to enhance the credibility and generalizability of the method.
2. Provide more detailed analysis of the results. The current presentation of results, especially on standard CL benchmarks like Split MNIST and CIFAR-100, is limited to numerical tables without sufficient discussion. Please include in-depth analysis of the performance, comparison across methods and failure cases to help readers better understand the method’s behavior.
3. Include more recent and stronger baselines. Most of the compared baselines are outdated.  Incorporating more recent continual learning methods would strengthen the empirical comparison and better position your contribution within the current state of the field.
4. Some entries in Table 1 are incomplete. If certain methods could not be run for specific settings, please explicitly state the reason.
5. An ablation study is necessary to evaluate the contribution of key components of your proposed framework. Please provide such an analysis to justify the design choices.
6. Please enhance the clarity and readability of the figures. For example, in Figure 1(b), it is difficult to distinguish instances from different classes. There are also seemingly misplaced texts in that figure.

**Strengths And Weaknesses:**

## Strengths

1. The framework presented in the paper is novel, which frames continual learning as inference over exchangeable sequences.
2. The method is task-agnostic and replay-free.
3. The paper is clearly written and provides strong theoritical analysis for the proposed approach.

## Weakness
1. The experimental evaluation is not comprehensive. Most experiments are conducted on two biometric datasets that are uncommon in the continual learning literature. Evaluation on standard CL benchmarks is limited to only MNIST and CIFAR-100, which weakens the generalizability claims. Moreover, a substantial portion of the manuscript in the experiment section is occupied by large figures and tables, some of which could be compressed or relocated to the appendix to improve clarity and space efficiency.
2. The compared methods are quite old. On their biometric datasets, the authors only compare with 3 baselines published before 2018. This is insufficient for a 2025 submission and does not convincingly demonstrate the method’s advantage over recent state-of-the-art approaches.
3. The empirical results are not particularly strong, and the experimental analysis is relatively shallow. More detailed discussion, interpretation, and ablation studies would strengthen the empirical contribution.

---

> ### Author Response · Authors · 2025-07-08
> **Responses to Reviewer SE8t (1)**
>
> The authors thank the reviewer for the constructive comments. We address the reviewer's concerns/requested changes.
>
> *  __Shifting the focus from biometric datasets to widely-used benchmarks__: We have revised the benchmarking section. We evaluated the proposed method on four widely-used public datasets: incremental CIFAR10 (iCIFAR-10), incremental CIFAR100 (iCIFAR-100), sequential Tiny ImageNet200 (S-TinyImageNet) and MNIST. We found that CL-BRUNO either outperforms existing exemplar-free methods or achieves competitive performance across all experimental settings. Details can be found in the revised text Sec 5.1 (_Changes are highlighted in blue._) We chose to keep the analysis of the two tabular biometric datasets in the main text because our work is motivated by biomedical applications, and we aim to introduce current continual learning (CL) methodologies to a broader interdisciplinary audience, particularly those from medical science and bioinformatics. We expect that the inclusion of non-conventional datasets could contribute to the CL research community by providing alternative evaluation scenarios and by stimulating research motivated by a wider range of scientific problems. Unfortunately we are not able to perform comprehensive benchmarking on the two biomedical datasets as we noticed that most of the current CL methods are either designed specifically for vision tasks or have vision-specific modules such as convolution layers integrated into their implementation. As a result, we are not able to directly apply them to tabular data without substantial architectural modification. This lack of attention to tabular data has also been discussed in recently published paper (Garcia-Santaclara et al., 2025). However we are not able to incorporate their method into the paper as the authors did not publish the implementation of their method.
>
> * __Provide more detailed analysis of the results.__: In addition to numerical tables, we have included analysis and comparison on memory overhead and prediction calibration. We found that CL-BRUNO's memory overhead is on a similar level to existing exemplar-free methods, and is able to achieve good prediction calibration in terms of Expected Calibration Error (Guo et al., 2017). Details can be found in the revised text Sec 5.
>
> * __Include more recent and stronger baselines.__: We compared our method with a wider range of baselines including CSReL-LODE (Tong et al., 2025), ReReLRP (Bogacka et al., 2025), NPCL (Jha et al., 2024), oEWC-MACL (Wang and Huang, 2024), NICE (Gurbuz et al., 2024), DS-AL (Zhuang et al., 2024), FeTRIL (Petit et al., 2023), SSRE (Zhu et al., 2022), Boo-VAE (Egorov et al., 2021), DER (Buzzega et al., 2020), PRER (Scardapane et al., 2020), ER (Riemer et al., 2018), MeRGAN (Wu et al., 2018), EWC (Kirkpatrick et al., 2017) and LwF (Li and Hoiem, 2017). Details can be found in the revised text Sec 5.
>
> * __Incomplete Table 1__: We have updated Table 1 and its caption in the revised text. Methods not applicable to the problem settings are now clearly labeled.
>
> * __Ablation study__: We investigate the contribution of the functional regulariser $R$ and the impact of pseudo data size $N'$. In main text we demonstrated how the distributional regulariser $L'$ in the loss function of CL-BRUNO naturally arises from the Bayesian updating rule. However, the inclusion of the functional regulariser $R$ is not justified. Therefore, we first investigate the role of the functional regulariser $R$ in the training of CL-BRUNO. We found that incorporating the functional regulariser $R$ in the training process of CL-BRUNO improves its performance by a large margin across all experiment setting. This confirms the effect of the functional regulariser $R$. Additionally, we also investigated the size of pseudo data $N'$ using the iCIFAR100 dataset. We run CL-BRUNO under four choices of pseudo samples $N'_1= 32, N'_2 = 64, N'_3 = 128$ and $N'_4=256$. The final classification accuracy of the four models is 10.9\%, 17.1\%, 21.2\% and 22.8\%, respectively, showing that increasing $N'$ leads to a diminishing increase in classification accuracy. We further investigate the impact of $N'$ on the computational cost. We found that increasing $N'$ by a factor of 8 (from 32 to 256) leads to a ~40\% increase in the running time of CL-BRUNO (from ~2500s to ~3500s). This confirms that increasing $N'$ does not drastically increase the computational cost of CL-BRUNO. Details can be found in the revised text Sec 5.
>
> * __Clarity and readability of the figures__: We have updated and rearranged the figures and their captions in the revised text. Figures are now more compactly arranged.

---

> > ### Author Response · Authors · 2025-07-08
> > **Responses to Reviewer SE8t (2)**
> >
> > * __Broader Impact Concerns__: We have incorporated a Broader Impact Statement in the revised text highlighting the scope and limitations of the proposed method. We acknowledge that generative models trained on real data can encode and reproduce distributional and/or identifiable patterns of the training data. One potential solution to this issue is to incorporate pretrained feature extractors in a similar fashion to the image datasets. By doing so, the model would only remember
> > the distributional patterns of the feature vectors instead of the original data, hence preventing the model from
> > reproducing the original training data. Although our experiments do not involve user-specific or sensitive
> > biometric data (PANCAN and ICI datasets are both non-identifiable data sets made available for research),
> > we recognise that applying such models to real-world, privacy-critical domains would necessitate more rigorous
> > safeguards. Future work should incorporate privacy-preserving techniques, e.g. differential privacy, to ensure
> > that generated samples do not leak sensitive information.
> > We also recognise the potential for bias amplification in continual learning settings, particularly when using
> > biometric datasets that may exhibit class imbalance or uneven demographic representation. Although
> > our current study focuses on methodological contributions rather than fairness-specific evaluations, we
> > believe that integrating fairness-aware strategies is essential for real-world deployment. Addressing these
> > ethical considerations is a key direction for future research, particularly for applications involving socially or
> > demographically sensitive data.

---

> > > ### Author Response · Authors · 2025-07-08
> > > **Responses to Reviewer SE8t (3)**
> > >
> > > _Bibliography_
> > >
> > > (Garcia-Santaclara et al., 2025)  Garcia-Santaclara, P., Fernandez-Castro, B., and Diaz-Redondo, R. P. (2025). Overcoming catastrophic forgetting in tabular data classification: A pseudorehearsal-based approach. Engineering Applications of Artificial Intelligence, 156:110908.
> > >
> > > (Tong et al., 2025) Tong, R., Liu, Y., Shi, J. Q., and Gong, D. (2025). Coreset selection via reducible loss in continual learning. In The Thirteenth International Conference on Learning Representations.
> > >
> > > (Bogacka et al., 2025) Bogacka, K., Höfler, M., Ganzha, M., Samek, W., and Wasielewska-Michniewska, K. (2025). Rerelrp–remembering and recognizing tasks with LRP. arXiv preprint arXiv:2502.10789.
> > >
> > > (Jha et al., 2024) Jha, S., Gong, D., Zhao, H., and Yao, L. (2024). Npcl: Neural processes for uncertainty-aware continual learning. Advances in Neural Information Processing Systems, 36
> > >
> > > (Wang and Huang, 2024) Wang, Z. and Huang, H. (2024). Model sensitivity aware continual learning. Advances in Neural Information Processing Systems, 37:132583–132613.
> > >
> > > (Gurbuz et al., 2024) Gurbuz, M. B., Moorman, J. M., and Dovrolis, C. (2024). Nice: Neurogenesis inspired contextual encoding
> > > for replay-free class incremental learning. In Proceedings of the IEEE/CVF Conference on Computer Vision and Pattern Recognition, pages 23659–23669.
> > >
> > > (Zhuang et al., 2024) Zhuang, H., He, R., Tong, K., Zeng, Z., Chen, C., and Lin, Z. (2024). Ds-al: A dual-stream analytic
> > > learning for exemplar-free class-incremental learning. In Proceedings of the AAAI Conference on Artificial Intelligence, volume 38, pages 17237–17244.
> > >
> > > (Petit et al., 2023) Petit, G., Popescu, A., Schindler, H., Picard, D., and Delezoide, B. (2023). Fetril: Feature translation for
> > > exemplar-free class-incremental learning. In Proceedings of the IEEE/CVF winter conference on applications of computer vision, pages 3911–3920.
> > >
> > > (Zhu et al., 2022) Zhu, K., Zhai, W., Cao, Y., Luo, J., and Zha, Z.-J. (2022). Self-sustaining representation expansion for
> > > non-exemplar class-incremental learning. In Proceedings of the IEEE/CVF Conference on Computer Vision and Pattern Recognition, pages 9296–9305.
> > >
> > > (Egorov et al., 2021) Egorov, E., Kuzina, A., and Burnaev, E. (2021). Boovae: Boosting approach for continual learning of vae. Advances in Neural Information Processing Systems, 34:17889–17901.
> > >
> > > (Buzzega et al., 2020) Buzzega, P., Boschini, M., Porrello, A., Abati, D., and Calderara, S. (2020). Dark experience for general
> > > continual learning: a strong, simple baseline. Advances in neural information processing systems, 33:15920–15930.
> > >
> > > (Scardapane et al., 2020) Scardapane, S., Uncini, A., et al. (2020). Pseudo-rehearsal for continual learning with normalizing flows. In 4th Lifelong Machine Learning Workshop at ICML 2020.
> > >
> > > (Riemer et al., 2018) Riemer, M., Cases, I., Ajemian, R., Liu, M., Rish, I., Tu, Y., and Tesauro, G. (2018). Learning to learn without forgetting by maximizing transfer and minimizing interference. arXiv preprint arXiv:1810.11910.
> > >
> > > (Wu et al., 2018) Wu, C., Herranz, L., Liu, X., Van De Weijer, J., Raducanu, B., et al. (2018). Memory replay gans: Learning to generate new categories without forgetting. Advances in neural information processing systems, 31.
> > >
> > > (Kirkpatrick et al., 2017) Kirkpatrick, J., Pascanu, R., Rabinowitz, N., Veness, J., Desjardins, G., Rusu, A. A., Milan, K., Quan, J.,
> > > Ramalho, T., Grabska-Barwinska, A., et al. (2017). Overcoming catastrophic forgetting in neural networks. Proceedings of the national academy of sciences, 114(13):3521–3526.
> > >
> > > (Li and Hoiem, 2017) Li, Z. and Hoiem, D. (2017). Learning without forgetting. IEEE transactions on pattern analysis and machine intelligence, 40(12):2935–2947.
> > >
> > > (Guo et al., 2017) Guo, C., Pleiss, G., Sun, Y., and Weinberger, K. Q. (2017). On calibration of modern neural networks. In
> > > International conference on machine learning, pages 1321–1330. PMLR.

---

> > > > ### Comment · Action_Editor_9XKT · 2025-08-17
> > > >
> > > > Dear reviewer SE8t,
> > > >
> > > > do the comments/explanations and updates in the paper resolve your concerns?
> > > > Are the additional experimental results (datasets and baseline methods) convincing enough? Or would you recommend adding some more? If so, can you please be concrete in proposing?
> > > > Are the ablations investigated by the authors those you had in mind or are there any others you would recommend as
> > > > particularly insightful?
> > > > Is the presentation of the results now improved enough to overall good readability or are there still any pending issues?
> > > > Is there anything else you would like to ask or discuss with the authors?
> > > >
> > > > Best,
> > > > your AE

---

> ### Comment · Reviewer_SE8t · 2025-08-18
>
> Thank the authors for the detailed response. Most of my concerns have been addressed. Good luck.

---

### Review · Reviewer_pAhH · 2025-06-01

**Summary Of Contributions:**

The authors propose CL-BRUNO, a CL method that leverages probabilistic, Neural Process-based models to address challenges such as catastrophic forgetting and uncertainty quantification in CL applications.

The method employs deep-generative models, specifically normalizing flows, to capture the joint density function of inputs and labels across a sequence of CL tasks, supporting generative replay without the need to store previously seen training examples. During CL, the model is iteratively trained using a combined objective, including  1) maximum likelihood on the union of training examples from the current task and synthetic examples from previous tasks through conditional generation, and 2) a functional regularization term to penalize discrepancies in the predictive outputs between the old and updated models. Inference is conducted by task-id prediction followed by label prediction, using the posterior task and label distributions.

The authors demonstrate the efficacy of CL-BRUNO through experiments on both standard CL benchmarks (MNIST and Split CIFAR-100) and biomedical datasets, showing that it outperforms baseline CL methods.

**Audience:**

Yes

**Claims And Evidence:**

No

**Requested Changes:**

Please kindly refer to the weaknesses mentioned above.

**Strengths And Weaknesses:**

### Strengths
- The paper conducts experiments in a medical application, a real-world continual learning scenario beyond synthetic benchmarks like MNIST, and demonstrates the good performance of the proposed method
- The proposed method does not retain previously seen samples, making it suitable for sensitive applications such as biomedical data analysis.
- The proposed method provides a unified probabilistic framework that can handle both task- and class-incremental CL problems.

### Weaknesses
- I think the paper lacks a thorough comparison with baseline methods. For example, the work by Scardapane et al. (2020) (also mentioned by the authors in the related work section), which employs normalizing flow for generative replay in continual learning, is not included in the comparisons. While the authors claim that their proposed method is more unified and scalable in the related work section, this is less convincing without concrete supporting evidence. Furthermore, the baseline results presented in Table 1 appear to be lower than those reported in the original publications. For instance, NPCL is reported at 0.198 by the authors, whereas the original paper reports a value of 0.3743. Is there a difference in the experimental setup that could account for this discrepancy?
- The paper does not include a systematic ablation study to explain the effectiveness of the proposed components. This omission limits the insights into why the proposed method would excel in performance compared to previous methods.
- The authors claim that uncertainty quantification is a weakness of existing methods and a strength of the proposed method. However, the experimental section does not adequately demonstrate how the proposed method improves in this area. There is no clear evidence or results related to uncertainty quantification of the proposed method, e.g., whether the method is better calibrated.
- The equation in the Method section on page 3 requires some clarification, particularly , the RHS because the right-hand side would include the term X_{1:0,t} when i =1 which can be confusing.

---

> ### Author Response · Authors · 2025-07-08
> **Responses to Reviewer pAhH (1)**
>
> The authors thank the reviewer for the constructive comments. We address the reviewer's concerns/requested changes.
>
> * __Thorough comparison with baseline methods__: We have revised the benchmarking section. We evaluated the proposed method on four widely-used public datasets: incremental CIFAR10 (iCIFAR-10), incremental CIFAR100 (iCIFAR-100), sequential Tiny ImageNet200 (S-TinyImageNet) and MNIST. We compared our method with a wider range of baselines including CSReL-LODE (Tong et al., 2025), ReReLRP (Bogacka et al., 2025), NPCL (Jha et al., 2024), oEWC-MACL (Wang and Huang, 2024), NICE (Gurbuz et al., 2024), DS-AL (Zhuang et al., 2024), FeTRIL (Petit et al., 2023), SSRE (Zhu et al., 2022), Boo-VAE (Egorov et al., 2021), DER (Buzzega et al., 2020), PRER (Scardapane et al., 2020), ER (Riemer et al., 2018), MeRGAN (Wu et al., 2018), EWC (Kirkpatrick et al., 2017) and LwF (Li and Hoiem, 2017). We found that CL-BRUNO either outperforms existing exemplar-free methods or achieves competitive performance across all experimental settings. Details can be found in the revised text Sec 5.1 (_Changes are highlighted in blue._) Regarding the results of NPCL: The authors of NPCL reported classification accuracy 37.4% and 15.3% on the iCIFAR100 and S-TinyImagenet dataset. Our independent run using the publicly available implementation provided by the authors under the same setup gave classification accuracies of 19.8% and 13.5% respectively. This could be due to environment differences. Our code for reproducing the NPCL results on iCIFAR100 and S-TinyImagenet can be found in Anonymous GitHub https://anonymous.4open.science/r/reproducing-2BE8.
>
> * __Ablation study__: We investigate the contribution of the functional regulariser $R$ and the impact of pseudo data size $N'$. In main text we demonstrated how the distributional regulariser $L'$ in the loss function of CL-BRUNO naturally arises from the Bayesian updating rule. However, the inclusion of the functional regulariser $R$ is not justified. Therefore, we first investigate the role of the functional regulariser $R$ in the training of CL-BRUNO. We found that incorporating the functional regulariser $R$ in the training process of CL-BRUNO improves its performance by a large margin across all experiment setting. This confirms the effect of the functional regulariser $R$. Additionally, we also investigated the size of pseudo data $N'$ using the iCIFAR100 dataset. We run CL-BRUNO under four choices of pseudo samples $N'_1= 32, N'_2 = 64, N'_3 = 128$ and $N'_4=256$. The final classification accuracy of the four models is 10.9\%, 17.1\%, 21.2\% and 22.8\%, respectively, showing that increasing $N'$ leads to a diminishing increase in classification accuracy. We further investigate the impact of $N'$ on the computational cost. We found that increasing $N'$ by a factor of 8 (from 32 to 256) leads to a ~40\% increase in the running time of CL-BRUNO (from ~2500s to ~3500s).  Details can be found in the revised text Sec 5.
>
> * __Uncertainty quantification__: The output of CL-BRUNO can be interpreted as the posterior distribution of labels given by an approximate Bayesian classifier, which in principle has a better probabilistic interpretation than the logit scores directly given by a neural network. To demonstrate CL-BRUNO's uncertainty quantification feature, we compare the Expected Calibration Error (ECE) (Guo et al., 2017)  of CL-BRUNO with both experience-replay and exemplar-free CL baselines in a similar fashion to Jha et al.(2024). We find that CL-BRUNO attains lower ECE than exemplar-free CL methods with similar accuracies (FeTriL and NICE), and show comparable performance to experience-replay methods on both iCIFAR10 and iCIFAR100 datasets. (See Table 2 in the revised text.) This suggests that the probabilistic nature of CL-BRUNO benefits it in terms of prediction calibration.
>    Additionally, we use the ICI dataset to show that CL-BRUNO is able to give probabilistic predictions that correctly reflects the uncertainty when tasks are indeed indistinguishable. Fig 2(c) and Fig3 show that the task identity predictions of Atezo samples are uncertain in a correct way that is inline with the underlying biological process  (Pal et al., 2022). This agreement between the uncertainty in probabilistic predictions given by CL-BRUNO and previous independent studies also demonstrates that CL-BRUNO is able to accurately quantify uncertainty from data.
>
> * __Equations in the Method section__: We have added clarification and updated the equations in the revised text. _Changes are highlighted in blue._

---

> > ### Author Response · Authors · 2025-07-08
> > **Responses to Reviewer pAhH (2)**
> >
> > _Bibliography_
> >
> > (Garcia-Santaclara et al., 2025)  Garcia-Santaclara, P., Fernandez-Castro, B., and Diaz-Redondo, R. P. (2025). Overcoming catastrophic forgetting in tabular data classification: A pseudorehearsal-based approach. Engineering Applications of Artificial Intelligence, 156:110908.
> >
> > (Tong et al., 2025) Tong, R., Liu, Y., Shi, J. Q., and Gong, D. (2025). Coreset selection via reducible loss in continual learning. In The Thirteenth International Conference on Learning Representations.
> >
> > (Bogacka et al., 2025) Bogacka, K., Höfler, M., Ganzha, M., Samek, W., and Wasielewska-Michniewska, K. (2025). Rerelrp–remembering and recognizing tasks with LRP. arXiv preprint arXiv:2502.10789.
> >
> > (Jha et al., 2024) Jha, S., Gong, D., Zhao, H., and Yao, L. (2024). Npcl: Neural processes for uncertainty-aware continual learning. Advances in Neural Information Processing Systems, 36
> >
> > (Wang and Huang, 2024) Wang, Z. and Huang, H. (2024). Model sensitivity aware continual learning. Advances in Neural Information Processing Systems, 37:132583–132613.
> >
> > (Gurbuz et al., 2024) Gurbuz, M. B., Moorman, J. M., and Dovrolis, C. (2024). Nice: Neurogenesis inspired contextual encoding
> > for replay-free class incremental learning. In Proceedings of the IEEE/CVF Conference on Computer Vision and Pattern Recognition, pages 23659–23669.
> >
> > (Zhuang et al., 2024) Zhuang, H., He, R., Tong, K., Zeng, Z., Chen, C., and Lin, Z. (2024). Ds-al: A dual-stream analytic
> > learning for exemplar-free class-incremental learning. In Proceedings of the AAAI Conference on Artificial Intelligence, volume 38, pages 17237–17244.
> >
> > (Petit et al., 2023) Petit, G., Popescu, A., Schindler, H., Picard, D., and Delezoide, B. (2023). Fetril: Feature translation for
> > exemplar-free class-incremental learning. In Proceedings of the IEEE/CVF winter conference on applications of computer vision, pages 3911–3920.
> >
> > (Zhu et al., 2022) Zhu, K., Zhai, W., Cao, Y., Luo, J., and Zha, Z.-J. (2022). Self-sustaining representation expansion for
> > non-exemplar class-incremental learning. In Proceedings of the IEEE/CVF Conference on Computer Vision and Pattern Recognition, pages 9296–9305.
> >
> > (Egorov et al., 2021) Egorov, E., Kuzina, A., and Burnaev, E. (2021). Boovae: Boosting approach for continual learning of vae. Advances in Neural Information Processing Systems, 34:17889–17901.
> >
> > (Buzzega et al., 2020) Buzzega, P., Boschini, M., Porrello, A., Abati, D., and Calderara, S. (2020). Dark experience for general
> > continual learning: a strong, simple baseline. Advances in neural information processing systems, 33:15920–15930.
> >
> > (Scardapane et al., 2020) Scardapane, S., Uncini, A., et al. (2020). Pseudo-rehearsal for continual learning with normalizing flows. In 4th Lifelong Machine Learning Workshop at ICML 2020.
> >
> > (Riemer et al., 2018) Riemer, M., Cases, I., Ajemian, R., Liu, M., Rish, I., Tu, Y., and Tesauro, G. (2018). Learning to learn without forgetting by maximizing transfer and minimizing interference. arXiv preprint arXiv:1810.11910.
> >
> > (Wu et al., 2018) Wu, C., Herranz, L., Liu, X., Van De Weijer, J., Raducanu, B., et al. (2018). Memory replay gans: Learning to generate new categories without forgetting. Advances in neural information processing systems, 31.
> >
> > (Kirkpatrick et al., 2017) Kirkpatrick, J., Pascanu, R., Rabinowitz, N., Veness, J., Desjardins, G., Rusu, A. A., Milan, K., Quan, J.,
> > Ramalho, T., Grabska-Barwinska, A., et al. (2017). Overcoming catastrophic forgetting in neural networks. Proceedings of the national academy of sciences, 114(13):3521–3526.
> >
> > (Li and Hoiem, 2017) Li, Z. and Hoiem, D. (2017). Learning without forgetting. IEEE transactions on pattern analysis and machine intelligence, 40(12):2935–2947.
> >
> > (Guo et al., 2017) Guo, C., Pleiss, G., Sun, Y., and Weinberger, K. Q. (2017). On calibration of modern neural networks. In
> > International conference on machine learning, pages 1321–1330. PMLR.
> >
> > (Pal et al., 2022) Pal, S. K., Uzzo, R., Karam, J. A., Master, V. A., Donskov, F., Suarez, C., Albiges, L., Rini, B., Tomita, Y.,
> > Kann, A. G., et al. (2022). Adjuvant atezolizumab versus placebo for patients with renal cell carcinoma at
> > increased risk of recurrence following resection (immotion010): a multicentre, randomised, double-blind,
> > phase 3 trial. The Lancet, 400(10358):1103–1116.

---

> > ### Comment · Action_Editor_9XKT · 2025-08-17
> > **Discussion with authors**
> >
> > Dear reviewer pAhH,
> > do the comments/explanations and updates in the paper resolve your concerns?
> > Are the ablations investigated by the authors those you had in mind or are there any others you would recommend as particularly insightful?
> > Do the explanations about the CL-BRUONO interpretation and the results of Table 2 sufficiently demonstrate the uncertainty quantification capabilities?
> > It there anything else you would like to ask or discuss with the authors?
> > Best,
> > your AE

---

> > > ### Comment · Reviewer_pAhH · 2025-09-03
> > > **Discussion with authors**
> > >
> > > Dear authors,
> > > Thank you for your detailed responses. My concerns regarding baselines, ablation studies and additional analysis have been addressed.

---

### Review · Reviewer_w7vW · 2025-08-14

**Summary Of Contributions:**

This paper adapts Conditional-BRUNO method to the continual learning setting (CL-BRUNO) and aims to demonstrate that it can perform competitively in a set of computer vision and biomedical CL datasets. More specifically, the C-BRUNO core allows generating pseudo-samples of a given task conditioned on a given label (using normalising flows), and these can then be used to mitigate forgetting of previously learned tasks. To further mitigate forgetting, the stability of the conditional generation is controlled by adding an L2 loss on the generated samples, to make sure they don’t differ much from the generated samples based on the previous state of the normalising flow.

The main contributions of the paper are:

* viewing CL under the lens of exchangeable sequence modelling
* proposing a probabilistic framework that allows unifying task-incremental learning and class-incremental learning

**Audience:**

Yes

**Broader Impact Concerns:**

None.

**Claims And Evidence:**

No

**Requested Changes:**

* Improve the motivation of the method in the CL context both in the abstract and the introduction (see comments above)
* Improve the positioning of CL-BRUNO relative to BRUNO/C-BRUNO. i.e. understanding that BRUNO/C-BRUNO are useful in contexts like few-shot learning can help. Also, the condition $0 < \rho < v$ is given without context, can you add a short explanation why it’s necessary in C-BRUNO? Maybe also add some extra short sentence on what the permutation/exchangeability means so that it’s easier to parse. Also notation in this Section is not consistent with the CL-BRUNO section (i.e. $h_i$ instead of $y_i$).
* Maybe adding a box with the algorithm would help the reader quickly know 1) how training happens, 2) how inference happens. It can also be a good place to clarify the process for getting $X_i$ samples, inferring task identity, and inferring $Y_i$.
* Improve quality/visibility of plots: at the moment the font size is too small.
* Bayesian CL literature seems to be missing, and this seems relevant given that one main motivation mentioned in the paper is the probabilistic nature of the method.
* At the moment it’s not clear to me why second paragraph in section 5.3 is interesting. These inter-task relationship results seem to come a bit out of nowhere, and take a lot of space in the main text. Can you add some explanation of why we care about these results, in the light of the specific method that you are proposing?
* Clarification: If indeed the default parameters were taken for the baselines, at least some hyperaparameter tuning seems necessary.
* Clarification: in 5.1 (training?) runtimes are given. In 5.2 you mention samples need to be generated for every gradient descent step. Can you explain why just in this experiment, and whether this has a significant impact on runtime?

Typos:

* 2.2: (as supposed to i.i.d.) → opposed
* 3.1: CL-BRUNO loss equation has no label that one can refer to, and in the last line of page 6, I think it should refer to that one and not to Eqn 6 as currently is
* 5.1: Ablation study: CL-RUNO_{dist}
* Figure 2 caption. ICI dataset consists of tasks with only one class - contains? As far as I understand, only some of them. As it’s currently written it seems all of them are.

**Strengths And Weaknesses:**

Strengths:

* The idea seems novel, interesting and relevant for handling CL. CL is in its core a problem about learning from non-i.i.d. data as the different tasks come in sequentially. In that context, viewing CL under the lens of exchangeable sequence modelling, which relaxes the i.i.d. assumption, seems pretty novel and interesting, and can provide a new set of tools to tackle sequential learning in a CL scenario.
* The paper is generally very nicely written, the notation is consistent, thorough and clear.

Weaknesses:

* A clear and strong motivation for the method in the CL context is missing. As I see, the main motivation is given at the end of 2.2 but this is very short and high level. Can you be more concrete in justifying how you are using the added flexibility of the exchangeability assumption in this CL setting, what it brings compared to other methods? In the abstract the main points for motivating the method are 1) not storing data and 2) having uncertainty quantification, but many other methods have already solved these problems. What else does this method bring? The abstract does not even mention exchangeability.
* The chosen experiments do not seem to demonstrate the advantages of this particular method. The paper highlights scalability (but is CIFAR100 really a dataset that demonstrates scalability?), ability to quantify uncertainty (but as I can see only the ECE table uses this, and there is no other mention to uncertainty in the results?), and the chosen datasets do not constitute cases where, within a given dataset, the exchangeability assumption is important? (e.g. the provided vision experiments to not need it, can you clarify if the biomedical experiments benefit from this somehow?).
* Certain statements are not supported by the reported results. E.g. Section 5.2. states “we see no abrupt interference or deterioration in prediction accuracy in the training steps”, this is a very vague statement and actually when looking at Fig 2(a) I would state the opposite. Same for 5.3 statement “Fig 2(b) show no drastic performance deterioration in any task”. There is clearly a drop in performance. Anyways these results are hard to interpret as there is no other baseline and no error bars. Also not sure if I understand correctly, but it seems the hyperparameters for the baselines were not tuned? If this is the case, it’s not a very fair comparison. Also, the paper mentions as motivation that TIL and CIL can be merged under a framework, but no experiments seem to demonstrate this (only potentially in appendix, but in a very ad-hoc synthetic dataset that is hard to judge, and that is never mentioned in the main).

---

> ### Author Response · Authors · 2025-08-27
> **Responses to Reviewer w7vW (1)**
>
> The authors thank the reviewer for the constructive comments. We address the reviewer's concerns/requested changes.
>
> * __Improve the motivation of the method__: We have revised the motivation paragraph. We clarified how the exchangeability assumption helps the model to better capture and aggregate distributional information such as sample sizes and inter-sample correlations in comparison with other generative-replay models based on the i.i.d. assumption (e.g. Scardapane et al. (2020)). In addition to the flexibility of C-BRUNO, we also highlight that by leveraging the efficient and exact density evaluation feature of C-BRUNO, our proposed method can give easy-to-interpret probabilistic predictions, which can be viewed as approximate Bayes classifiers, without the need of training or maintaining separate classifiers as in e.g.  Petit et al. (2023) and Wu et al. (2018). It simplifies the architecture of the proposed model, and improves its computation efficiency. Details can be found in revised text Sec 2.2 second paragraph highlighted in red.
>
> * __Improve the positioning of CL-BRUNO relative to BRUNO/C-BRUNO, the condition $0<\rho<\nu$ and notation__: The condition $0<\rho<\nu$ in C-BRUNO is to ensure that the resulting Multivariate Gaussian distribution is well-defined (i.e. forcing the covariance matrix to be positive definite). This is a special case of the requirement that a valid kernel function of a Gaussian process needs to be positive definite. We have included additional explanation on permutation invariance and exchangeability. We have also clarified how CL-BRUNO is built of C-BRUNO's modelling framework, and how we leverage features of C-BRUNO in our proposed model. We also revised the notation. Changes can be found in revised text Sec 2.2 highlighted in red.
>
> * __Algorithm box__: We have added an algorithm box summarising the training and inference procedures of CL-BRUNO for TIL problems in Sec 3.1. CL-BRUNO for CIL and their combination work similarly.
>
> * __Improve quality/visibility of plots__: We have updated the arrangement of Figure 1,2,3 in the revised text highlighted in red.
>
> * __Bayesian CL literature__: We added discussion on recent Bayesian CL methods in the revised Sec 4.1 highlighted in red.
>
> * __Why second paragraph in section 5.3 is interesting__: Some cancers evade detection by the immune system by emitting chemical signals called *immune checkpoint proteins* to turn the immune system off. Each of the seven treatments considered in the Molecular Response to Immune Checkpoint Inhibitors (ICI) dataset (Eddy et al., 2020) is specifically designed to block immune checkpoint proteins and therefore reactivate the anti-cancer activity of the immune system. For example, Atezolizumab (atezo) blocks a checkpoint protein called PD-L1 which is produced in some lung and skin cancers. The effectiveness of an ICI is dependent on whether the cancer produces the checkpoint protein the ICI is designed to block. In this experiment, we wanted to determine if CL-BRUNO could learn the molecular signatures which determine the effectiveness of each ICI which can be treated as a proxy cancer type classification task. Specifically, we would like to examine whether CL-BRUNO exhibits the correct probabilistic output behaviour. By investigating inter-task relationship in the second paragraph of Sec 5.3, we demonstrate that CL-BRUNO is able to give probabilistic predictions that correctly reflects the uncertainty when tasks are indeed indistinguishable. Fig 2(c) and Fig 3 show that the task identity predictions of Atezo samples are confused or inaccurate in a correct way that is inline with the underlying biological process (Pal et al., 2024). This agreement between the probabilistic predictions given by CL-BRUNO and previous independent studies demonstrates that CL-BRUNO is able to capture and quantify uncertainty accurately. We revised Sec 5.3 and highlighted this point.

---

> > ### Author Response · Authors · 2025-08-27
> > **Responses to Reviewer w7vW (2)**
> >
> > * __Hyperparameters of the baselines__: In Sec 5.1, our benchmarking settings on iCIFAR10, iCIFAR100 and TinyImagenet follow the experiment designs in Jha et al (2024). As a result, for NPCL and other methods adopting the same experiment designs such as DER, LwF, oEWC-MACL and NICE, the recommended hyperparameters reported by the authors have already been fine-tuned. Methods such as FeTriL and ReReLRP were also applied to the same datasets under very similar settings in their original papers. Hence, we also used hyperparameters reported in the original papers for these methods. We further experimented with ReReLRP by investigating the model performance under different hyperparameter values centred at the recommended ones. We found the results are not sensitive to the hyperparameter choices. Hence we believe we provided a fair comparison in Sec 5.1 without resorting to fine-tuning all previously published experiments. Regarding the two biomedical datasets, we would like to clarify that we are not able to perform comprehensive benchmarking on them as in Sec 5.1 since we noticed that most of the current CL methods are either *designed specifically for vision tasks* or *have vision-specific modules such as convolution layers* integrated into their implementations. As a result, we are not able to directly apply them to tabular data without substantial architectural modification. This lack of attention to tabular data has also been discussed in Garcia-Santaclara et al. (2025). However we are not able to include their method into the paper as the authors did not publish an implementation of their method. We also revised the text in Sec 5.2 and provided more accurate descriptions of the knowledge retention curves.
> >
> > * __Generating pseudo-samples for every gradient descent step__: We would like to clarify that generating pseudo data at each gradient descent step is a part of the training procedure of CL-BRUNO, and is used in all our examples. We have clarified it in the updated text Sec 5.1, 5.2. We investigated the computational cost of generating pseudo data using the iCIFAR100 dataset as an example. We found that increasing pseudo data size $N'$ by a factor of 8 (from 32 to 256) leads to a $\sim 40$% increase in the total training time of CL-BRUNO. This confirms that increasing $N'$ within a sensible range does not drastically increase the computational cost of CL-BRUNO. Details can be found in Sec 5.1 Ablation study.
> >
> > * __Typos__: Typos have been fixed and equations are now correctly labeled.
> >
> >
> > _Bibliography_
> >
> > (Scardapane et al., 2020) Scardapane, S., Uncini, A., et al. (2020). Pseudo-rehearsal for continual learning with normalizing flows. In 4th Lifelong Machine Learning Workshop at ICML 2020.
> >
> > (Petit et al., 2023) Petit, G., Popescu, A., Schindler, H., Picard, D., and Delezoide, B. (2023). Fetril: Feature translation for exemplar-free class-incremental learning. In Proceedings of the IEEE/CVF winter conference on applications of computer vision, pages 3911–3920.
> >
> > (Wu et al., 2018) Wu, C., Herranz, L., Liu, X., Van De Weijer, J., Raducanu, B., et al. (2018). Memory replay gans: Learning to generate new categories without forgetting. Advances in neural information processing systems, 31.
> >
> > (Eddy et al., 2020) Eddy, J. A., Thorsson, V., Lamb, A. E., Gibbs, D. L., Heimann, C., Yu, J. X., Chung, V., Chae, Y., Dang, K., Vincent, B. G., et al. (2020). Cri iatlas: an interactive portal for immuno-oncology research. F1000Research, 9.
> >
> > (Pal et al., 2022) Pal, S. K., Uzzo, R., Karam, J. A., Master, V. A., Donskov, F., Suarez, C.,
> > Albiges, L., Rini, B., Tomita, Y., Kann, A. G., et al. (2022). Adjuvant ate-
> > zolizumab versus placebo for patients with renal cell carcinoma at increased risk
> > of recurrence following resection (immotion010): a multicentre, randomised,
> > double-blind, phase 3 trial. The Lancet, 400(10358):1103–1116.
> >
> > (Jha et al., 2024) Jha, S., Gong, D., Zhao, H., and Yao, L. (2024). Npcl: Neural processes for uncertainty-aware continual learning. Advances in Neural Information Processing Systems, 36
> >
> > (Garcia-Santaclara et al., 2025)  Garcia-Santaclara, P., Fernandez-Castro, B., and Diaz-Redondo, R. P. (2025). Overcoming catastrophic forgetting in tabular data classification: A pseudorehearsal-based approach. Engineering Applications of Artificial Intelligence, 156:110908.

---

> > > ### Comment · Reviewer_w7vW · 2025-08-28
> > >
> > > I thank the authors for their answers and for addressing my concerns.

---

### Decision · Action_Editor_9XKT · 2025-09-28

**Recommendation:** Accept as is

**Audience:**

Yes

**Audience Explanation:**

The paper extends the previously existing method for C-BRUNO for conditional modelling of exchangeable sequences to the continual learning setting proposing a CL-BRUNO. The method - based on the normalizing flows - uses generative replay and functional regularization for class- and task-incremental learning. Though the introduced methods is incremental building on previously proposed approaches, it combines these in a novel way and shows in experiments, that it could be of interest to the community.

**Claims And Evidence:**

Yes

**Claims Explanation:**

During the review discussion reviewers raised several points of concern related to the motivation, positioning within the state of art and experimental evaluation. Authors have improved the paper in multiple aspects. In particular, they have made multiple updates to the experimental section to demonstrate the performance of their method more convincingly.